# Critical Tests of Leading Gamma Ray Burst Theories

**Shlomo Dado [1,†]** , **Arnon Dar [1,*,†]** **and Alvaro De Rújula [2,3,†]**

1 Department of Physics, Technion, Israel Institute of Technology, Haifa 32000, Israel; dado@phep3.technion.ac.il
2 Institute of Theoretical Physics, Autonomous University of Madrid, 28049 Madrid, Spain; alvaro.derujula@cern.ch
3 The European Organization for Nuclear Research, 1211 Meyrin, Switzerland
* Correspondence: arnon@physics.technion.ac.il
† These authors contributed equally to this work.

**Abstract:** It has been observationally established that supernovae (SNe) of Type Ic produce long duration gamma-ray bursts (GRBs) and that neutron star mergers generate short hard GRBs. SN-Less GRBs presumably originate in a phase transition of a neutron star in a high mass X-ray binary. How these phenomena actually generate GRBs is debated. The fireball and cannonball models of GRBs and their afterglows have been widely confronted with the huge observational data, with their defenders claiming success. The claims, however, may reflect multiple choices and the use of many adjustable parameters, rather than the validity of the models. Only a confrontation of key falsifiable predictions of the models with solid observational data can test their validity. Such critical tests are reviewed in this report.

**Keywords:** gamma ray bursts; afterglows; supernovae; neutron star mergers; cannonballs; fireballs





## 1. Introduction

Gamma-ray bursts (GRBs) are brief flashes of gamma rays lasting between a few milliseconds and several hours, message-bearers of extremely energetic astrophysical phenomena [1]. They were first observed on 2 July 1967 by the USA Vela spy satellites, which were launched to detect USSR tests of nuclear weapons in the atmosphere, in violation of the 1963 USA-USSR Nuclear Test Ban Treaty. Their discovery was first published in 1973 after 15 such events were detected [2], establishing their "natural" character and indicating an extra-solar origin.

During the first 20 years after their discovery, hundreds of GRB models were published, all assuming that GRBs were Galactic in origin (see, e.g., [3]). An extragalactic origin would imply an implausibly large energy release in gamma rays from a very small volume in a very short time, if their emission was isotropic, as was generally assumed. During that period it was also found that GRBs fall roughly into two classes, long duration ones (LGRBs) that last more than $\sim$2 s, and short bursts (SGRBs) lasting less than $\sim$2 s [4,5]. Most SGRBs are short hard bursts (SHBs), with a much harder spectrum than LGRBs. The origin and production mechanism of GRBs have been major astrophysical puzzles until recently.

In 1984, Blinnikov et al. [6] suggested that exploding neutron stars in close binaries may produce GRBs with an isotropic gamma-ray energy up to $\sim$$10^{46}$ erg. Such GRBs could be seen only from relatively nearby galaxies. Per contra, Paczynski maintained [7] that the sky distribution of GRBs was more consistent with large cosmological distances, such as those of quasars, with a typical redshift, $z$, between 1 and 2. This would imply a supernova-like energy release, $\sim 10^{51}$ erg, within seconds, making gamma-ray bursters the brightest known objects, many orders of magnitude brighter than any quasar [7].

The first plausible physical model of GRBs at large cosmological distances was proposed by Goodman, Dar and Nussinov in 1987 [8]. They suggested that GRBs were produced in stripped-envelope SNe and neutron stars mergers (NSMs) by an $e^+e^-\gamma$ fireball [9]

formed by neutrino-antineutrino annihilation around the newly born compact object—a massive neutron star, a quark star or a stellar black hole. Shortly after the launch of the Compton Gamma-Ray Observatory (CGRO) in 1991, it became clear that neutrino-annihilation fireballs were not powerful enough to produce observable GRBs at the cosmological distances indicated by the CGRO observations [10], unless the fireballs were collimated into narrow beams by funneling through surrounding matter, as posited by Meszaros and Rees [11].

In 1994, Shaviv and Dar suggested [12] that narrowly beamed GRBs can be produced by jets of highly relativistic plasmoids of ordinary matter, later called *cannonballs* (CBs), via the inverse Compton scattering (ICS) of light surrounding their launch sites. They proposed that such jets may be ejected in stripped-envelope core-collapse supernova explosions, in mergers of compact stars due to the emission of gravitational waves, and in a phase transitions of neutron stars (NSs) to an even more compact object, i.e., a quark star or a black hole (BH), following mass accretion in compact binaries. These hypotheses, unaltered, constitute the basis of the CB model.

An important prediction of the fireball (FB) model was a transition of the initial short $\gamma$-ray emission to a longer-lived *afterglow* (AG) [13–15] at longer wavelengths, due to the slow down of the expansion of the $e^+e^-\gamma$ fireball by the surrounding medium. In 1997 the team behind the satellite BeppoSAX discovered that GRBs are indeed followed by a longer-lived X-ray AG [16]. This resulted in an accurate enough sky localization of GRBs, facilitating the discovery of their AGs at longer wavelengths [17,18], the localization of their host galaxies [19] and the measurement of their redshifts [20] and, at least in one early instance, the association of a GRB with a type Ic SN explosion [21].

During the past twenty years, observers mainly using HETE, Swift, Konus-Wind, Chandra, Integral, XMM-Newton, Fermi and the Hubble space telescope, plus ground-based telescopes, measured the spectra of GRBs from $\gamma$-rays to radio. They demonstrated the association of LGRBs with Type Ic SNe and studied the properties of their host galaxies and near environments [22]. In particular, this provided clear evidence that LGRBs take place mainly in star formation regions within the disk of spiral galaxies, where most Type Ic SNe take place, while SGRBs originate in and outside spiral and elliptical galaxies and are not associated with SN explosions. These differences led to the wide spread recent belief [23,24] that SHBs are produced in mergers of two NSs, or a NS and a BH, as suggested long before [8,11].

The Ligo–Virgo gravitational wave (GW) detectors made it possible to test whether relatively nearby mergers produce SGRBs. Indeed, SHB170817A, ref. [25–29] was seen $1.74 \pm 0.05$ s after the gravitational wave burst GW170817 [30–33], proving that NSMs produce SHBs. Moreover, the universal shape of all the well sampled early AGs of ordinary SHBs and of SHB170817A—expected from a pulsar wind nebula (PWN) powered by the spin down of a newly born millisecond pulsar—suggests that most SHBs are produced by NSMs yielding an NS remnant rather than a black hole [34].

Although LGRBs have been seen in association with type Ic SNe [21,35,36], no associated SN has been detected in several nearby long duration GRBs, despite very deep searches [37,38]. The universal behavior of the AG of both SHBs and SN-Less GRBs [34,39] suggests that the latter are also powered by a newly born millisecond pulsar, perhaps after a phase transitions of an NS to a quark star [12,40], following mass accretion onto the NS in a high mass X-ray binary (HMXRB).

Since 1997 only two theoretical models of GRBs and their afterglows—the standard fireball (FB) model [24,41–52] and the cannonball (CB) model [12,53–57]—have been used extensively to interpret the mounting observations. Practitioners of both models have claimed to reproduce well the data. But the two models were originally and still are quite different in their basic assumptions and predictions. This is despite the replacements, to be mentioned, of key assumptions of the standard FB model (but not its name) with assumptions underlying the CB model. The claimed success of the models in reproducing the data, despite their complexity and diversity, may reflect the fact that most theoretical

results depend on free parameters and choices adjusted for each GRB. As a result, when successful fits to data were obtained, it was not clear whether they were due to the validity of the theory or to multiple choices and the use of many adjustable parameters.

Scientific theories ought to be falsifiable. Hence, only confrontations between solid observational data and key predictions of theories, which do not depend on free adjustable parameters, can serve as decisive tests of the theories. Such critical tests of the cannonball and fireball models of long GRBs and SHBs are summarized in this review.

## 2. The GRB Models

GRBs and SHBs consist of a few $\gamma$-ray pulses with a "FRED" temporal shape: a fast rise and an (approximately) exponential decay [1]. The number of pulses, their chaotic time sequence and their relative intensities vary drastically from burst to burst and are not predicted by the GRB models. The main properties of resolved pulses and of the AGs of GRBs and SHBs, as well as the correlations between different observables are what the models ought to predict or understand. Since LGRBs and SHBs have different progenitors, they will be discussed separately.

### 2.1. The Cannonball Model

The CB model [12,53–57] is illustrated in Figure 1. In it, bipolar jets of highly relativistic plasmoids (a.k.a. CBs) are assumed to be launched by matter falling onto a newly born compact stellar object [12]. SNe of Type Ic (the broad-line stripped-envelope ones) thus generate "SN-GRBs". Similarly, mergers in NS/NS and NS/BH binaries give rise to SHBs. SN-less GRBs are produced in high-mass X-ray binaries, as an NS accreting mass from a companion suffers a phase transition to a quark star [12]. Finally X-ray flashes (XRFs) are simply GRBs observed from a relatively large angle relative to the CBs' emission axis.

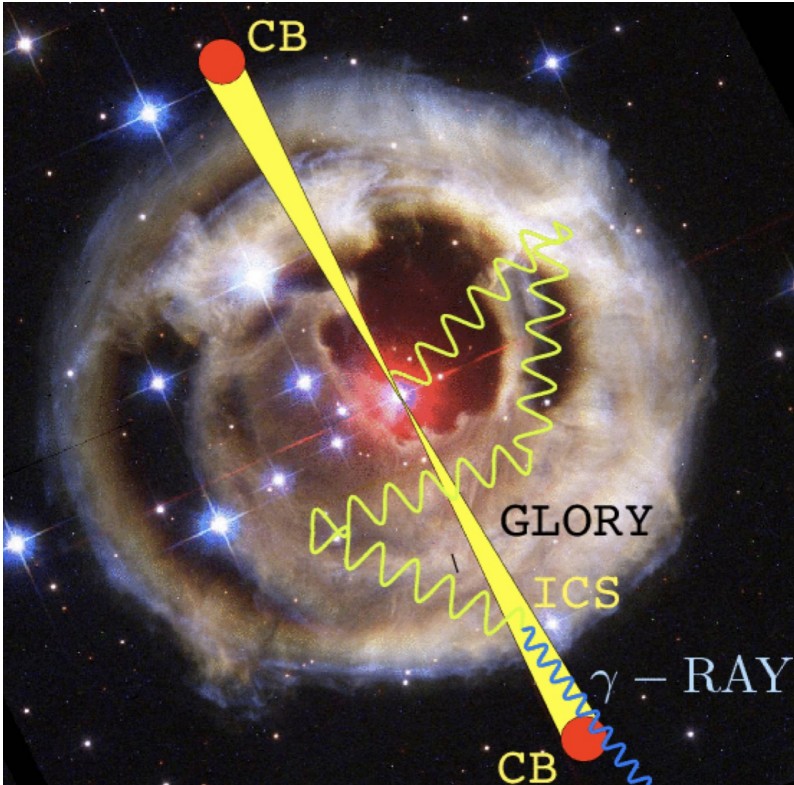

**Figure 1.** Electrons in a cannonball inverse Compton scatter photons in the glory of light surrounding a newly-born compact object, launching them forward as a narrow beam of $\gamma$ rays.

The prompt $\gamma$-ray pulses are produced by ICS by the electrons enclosed in the CBs of the radiation surrounding the launch site: the "glory". In SN-GRBs, this glory is the light halo formed around the progenitor star by scattered light from pre-supernova ejections [12,53–57]. In SN-less GRBs the glory can be light from the massive star companion, or the radiation emitted from the accretion disk formed around the NS. In SHBs it can be the X-ray radiation from an accretion disk formed around the NS remnant by fall back of tidally disrupted material or debris from the final explosion of the lighter NS after it lost most of its mass [6].

The prompt radiation is more than intense enough to completely ionize the interstellar medium (ISM) of the host galaxy along the path that the CBs will follow. When a CB enters this medium, it decelerates by sweeping in the nuclei and electrons in front of it. The swept-in particles are Fermi accelerated to high energies by the turbulent magnetic fields present or generated in the CBs by the merging inner and interstellar plasmas. The accelerated electrons emit synchrotron radiation (SR), the dominant AG of SN-GRBs, that usually take place in dense stellar regions: the molecular clouds where most SNe occur.

In SN-less GRBs and SHBs with a millisecond pulsar remnant, which usually take place in much lower density environments than those of SN-GRBs, the AG appears to be dominated by the radiation emitted from the pulsar wind nebula (PWN) [39,40].

*2.2. The Fireball Model*

The FB models of GRBs evolved a long way from the original spherical $e^+e^-\gamma$ fireball [9] to the current "collimated-fireball" models [24,41–52]. A very popular version is illustrated in Figure 2. Long GRBs are produced by a jet of highly relativistic conical shells of ordinary matter launched by *collapsars*—the collapse of a massive star to a black hole—either directly without a supernova (*failed* supernova) [58,59], or indirectly in a *hypernova*: the delayed collapse of the newly-born compact object to a BH by accretion of fall back material in a core-collapse SN [60,61]. SHBs are assumed to be produced by similar jets, launched in the merger of an NS with another one, or with a BH.

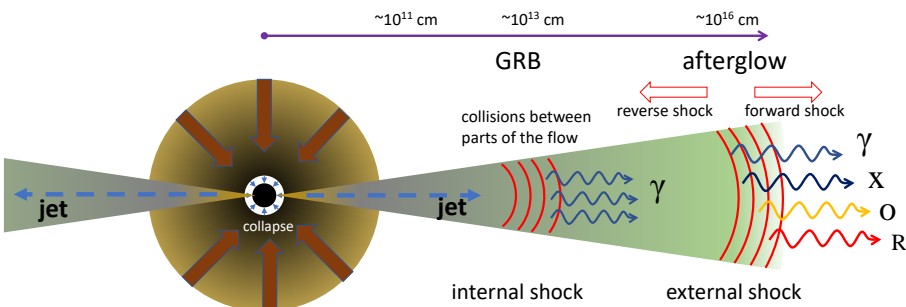

**Figure 2.** Schematic description of the fireball model of GRBs.

In the current FB models, the prompt emission pulses are assumed to be produced by synchrotron radiation (SR) from highly relativistic electrons shock-accelerated in the collisions between colliding conical shells. The collision of the shells with the circumburst medium is assumed to drive a forward shock into this medium or the pre-ejected stellar wind, and a reverse shock towards the inner shells. The electrons produce the AG [24,41–52] on top of the light of a hypernova [61] in LGRBs, or a *macronova* in SHBs [6,62]. The reverse shock produces the optical photons while ICS of the SR in the forward blast wave produces photons of GeV to TeV energy.

## 3. The Prompt GRB Emission

The CB-model's basic assumption is that, in analogy with quasars and micro-quasars, a SNIc event results in the axial emission of opposite jets of one or more CBs, made of ordinary matter. The $\gamma$-rays of a GRB are produced by jets of CBs with an initial bulk-motion Lorentz factor $\gamma_0 \equiv \gamma(t=0) \gg 1$. Although initially expanding in their rest system

at a speed of the order of the relativistic sound speed, $c/\sqrt{3}$, CBs are effectively point-like for an earthly observer at a cosmological distance.

The electrons in a CB inverse-Compton-scatter the ambient photons they encounter. This results in a *prompt* $\gamma$-ray beam of aperture $\simeq 1/\gamma_0 \ll 1$ around the CB's direction. Viewed by an observer at an angle $\theta$ relative to the CB's direction, the individual photons are boosted in energy by a Doppler factor $\delta_0 \equiv \delta(t=0) = 1/[\gamma_0 (1 - \beta \cos\theta)]$ or, to a good approximation for $\gamma_0^2 \gg 1$ and $\theta^2 \ll 1$, $\delta_0 \simeq 2\gamma_0/(1 + \gamma_0^2 \theta^2)$.

### 3.1. The GRB Polarization (Test 1)

For the most probable viewing angles ($\theta \sim 1/\gamma_0$) the polarization of the prompt, inverse Compton scattered photons is linear and its predicted [12,54] magnitude is:

$$\Pi = 2\gamma_0^2 \theta^2 / (1 + \gamma_0^4 \theta^4) = \mathcal{O}(1). \tag{1}$$

Very luminous or very dim GRBs are likely, respectively, to have been observed very near ($\gamma_0^2 \theta^2 \ll 1$) or very far off-axis ($\gamma_0^2 \theta^2 \gg 1$), thus resulting in considerable but not near-to-maximal polarization.

In the standard FB models, both the prompt GRB and the AG are produced by SR from high energy electrons, shock- and Fermi-accelerated in collisions between conical shells, and between conical shells and the ISM, respectively. Such acceleration requires highly turbulent magnetic fields in the acceleration region, resulting in a rather small net polarization. Indeed, the AGs are observed to display a small polarization [63] as predicted by both the CB and the FB models. But, while the polarization of the prompt emission in the FB models is expected to be small, such as that of the AG, a rather large linear polarization of the prompt emission has been observed in most GRBs where it was measured [64–72]. In Table 1 the available polarization data are reported. Though these measurements are challenging, the message that the polarization is large is loud and clear.

**Table 1.** GRBs with measured $\gamma$-ray polarization during the prompt emission.

| GRB | Polarization(%) | CL | Reference [64–72] | Polarimetry |
|---|---|---|---|---|
| 930131 | $>35$ | 90% | Willis et al., 2005 | BATSE (Albedo) |
| 960924 | $>50$ | 90% | Willis et al., 2005 | BATSE (Albedo) |
| 021206 | $80 \pm 20$ | ? | Coburn & Boggs, 2003 | RHESSI |
| 041219A | $98 \pm 33$ | 68% | Kalemci et al., 2007 | INTEGRAL-SPI |
| 100826A | $27 \pm 11$ | 99% | Yonetoku et al., 2011 | IKARUS-GAP |
| 110301A | $70 \pm 22$ | 68% | Yonetoku et al., 2012 | IKARUS-GAP |
| 110721 | $84 +16/-28$ | 68% | Yonetoku et al., 2012 | IKARUS-GAP |
| 061122 | $>60$ | 68% | Gotz et al., 2013 | INTEGRAL-IBIS |
| 140206A | $>48$ | 68% | Gotz et al., 2014 | INTEGRAL-IBIS |
| 160821A | $66 +27/-26$ | 99% | Sharma et al., 2019 | AstroSat-CZTI |
| 190530A | $55.4 \pm 21.3$ | 99% | Gupta et al., 2022 | AstroSat-CZTI |

### 3.2. Prompt-Observable Correlations (Test 2)

The ICS of glory photons of energy $\epsilon$ by a CB boosts their energy, as seen by an observer at redshift $z$, to $E_\gamma = \gamma_0 \delta_0 \epsilon / (1+z)$. Consequently, the peak energy $E_p$ of their time-integrated energy distribution satisfies:

$$(1 + z) E_p \approx \gamma_0 \delta_0 \epsilon_p, \tag{2}$$

with $\epsilon_p$ the peak energy of the glory.

In the Thomson regime the nearly isotropic distribution (in the CB rest frame) of a total number $n_\gamma$ of IC-scattered photons is beamed into an angular distribution $dn_\gamma/d\Omega \approx (n_\gamma/4\pi) \delta^2$ in the observer's frame. Consequently, the isotropic-equivalent total energy of the photons satisfies:

$$E_{iso} \propto \gamma_0 \, \delta_0^3 \, \epsilon_p. \tag{3}$$

Hence, both ordinary LGRBs and SGRBs, which in the CB model are GRBs viewed mostly from an angle $\theta \approx 1/\gamma$ (for which $\delta_0 \approx \gamma_0$), satisfy the correlation:

$$(1+z) \, E_p \propto [E_{iso}]^{1/2}, \tag{4}$$

while far off-axis ones $(\theta^2 \gg 1/\gamma^2)$ have a much lower $E_{iso}$, and satisfy

$$(1+z) \, E_p \propto [E_{iso}]^{1/3}. \tag{5}$$

These $[E_p, E_{iso}]$ correlations, predicted by the CB model [53,73,74], were later empirically discovered [75] in ordinary LGRBs. They are shown in Figures 3 and 4 for GRBs of known redshift. The $[E_p, E_{iso}]$ correlation predicted by the CB model for low luminosity SGRBs is presented in Figure 5. The figure includes SHB170817A, a very soft burst known for its association with a gravitational-wave signal [25–29]. This SHB will deserve its own chapter.

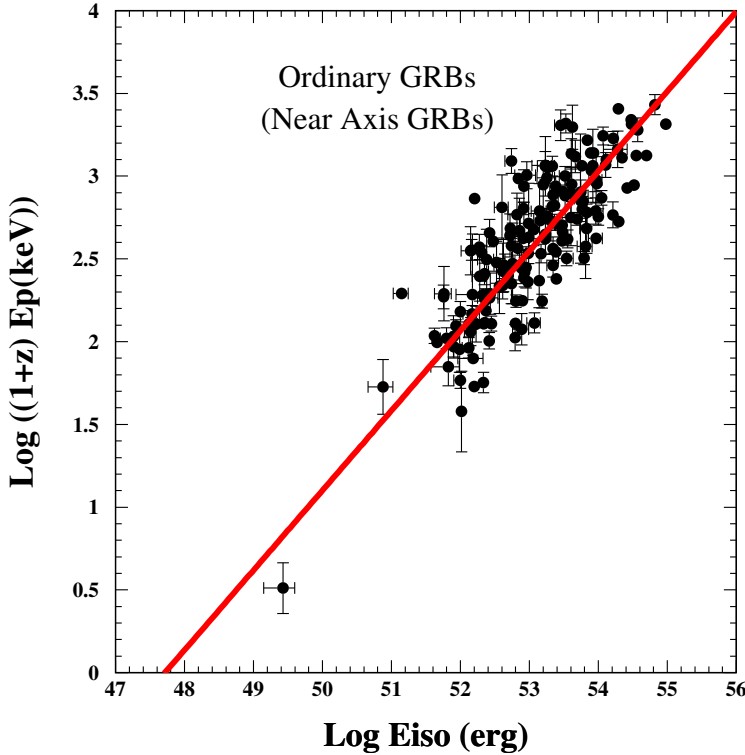

**Figure 3.** The $[E_p, E_{iso}]$ correlation in ordinary LGRBs viewed near axis. The line is the best fit, whose slope is $0.48 \pm 0.02$, consistent with the CB model prediction of Equation (4). The lowest $E_{iso}$ GRB is 020903 at $z = 0.25$ (HETE).

An apparent weakness of Equations (2) and (3) is that, strictly speaking, they refer to single-pulse GRBs: they are written for just one value of the Lorentz factor and of the viewing angle. Based on Fermi data, it has been shown that the $[E_p, E_{iso}]$ correlation is very well fit by $(1+z)E_p \propto E_{iso}^b$ for single or first pulses ($b = 0.465 \pm 0.044$; $\chi^2/\mathrm{dof} = 1.10$), for the rest of the pulses ($b = 0.503 \pm 0.050$; $\chi^2/\mathrm{dof} = 1.09$) and even for the entire GRB ($b = 0.499 \pm 0.035$; $\chi^2/\mathrm{dof} = 1.04$) [76]. These results are in excellent agreement with the CB-model prediction, Equation (4).

The correlations just discussed, snuggly satisfied by the data and extending over many orders of magnitude, strongly support the contention that the prompt photons of high and low luminosity GRBs—as well as SHBs and XRFs—are emitted by an effectively point-like highly relativistic source—such as a CB—viewed at different angles. The FB

models have not been shown to predict or explain the "Amati correlation", neither for GRBs, nor for SHBs.

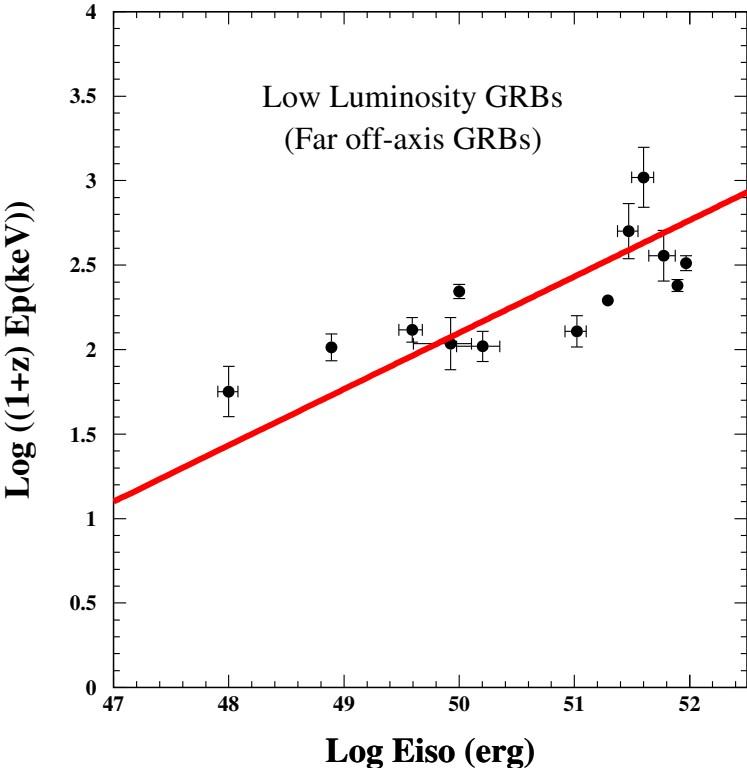

**Figure 4.** The $[E_p, E_{iso}]$ correlation in LGRBs viewed far off axis (which include the so-called low-luminosity LGRBs, and XRFs) [77]. The line is the CB model prediction [53,73,74] of Equation (5).

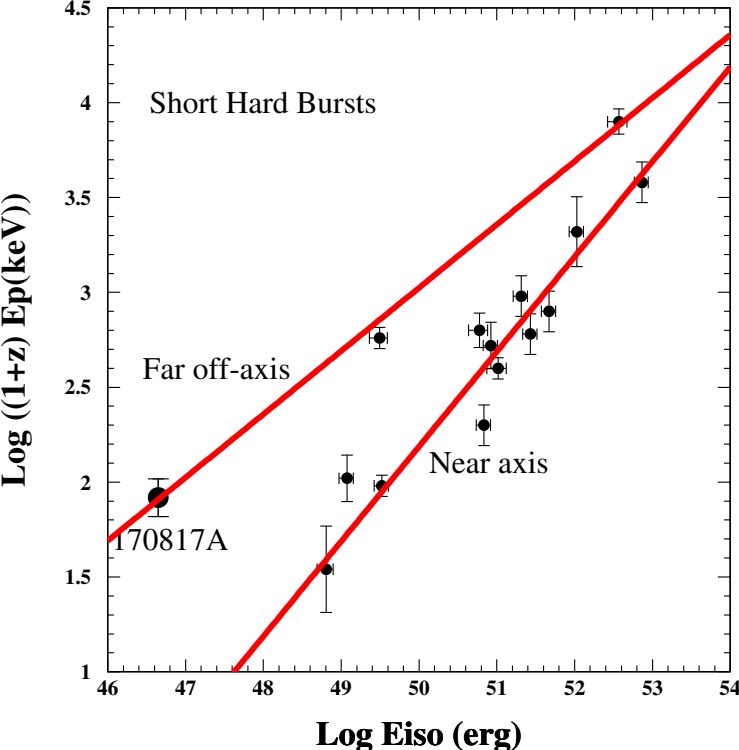

**Figure 5.** The $[E_p, E_{iso}]$ correlations in SHBs. The lines are the CB model predicted correlations as given by Equations (4) and (5).

### 3.3. Temporal Shape of Prompt Pulses (Test 3)

As stated before, GRBs consist of fast rising, roughly exponentially decaying individual short pulses [1], dubbed "FRED". Their number, time sequence, and relative intensities vary drastically between bursts and are not predictable by the current GRB models.

### 3.3.1. Pulse Shapes in the CB Model

The typical FRED shape of individual pulses is predicted in the CB model, and their durations are understood. The glory's light has a thin thermal bremsstrahlung spectrum $dn_g/d\epsilon \propto \epsilon^{-\alpha} \exp[-\epsilon/\epsilon_p]$ [78]. Inverse Compton Scattering of this light by the electrons in a CB produces a GRB pulse with a time and energy dependence well described [79] by:

$$E \frac{d^2 N_\gamma}{dE\,dt} \propto \frac{t^2}{(t^2 + \Delta^2)^2} \, E^{1-\alpha} \, \exp[-E/\mathcal{E}_p(t)] \tag{6}$$

where $\mathcal{E}_p(t)$ is the pulse's peak energy at time $t$, $E_p \equiv \mathcal{E}_p(t=0)$ is its maximum, $\Delta$ is approximately the peak time of the pulse in the observer's frame—originating at the time when the CB becomes transparent to inverse Compton scattering.

In Equation (6), the early quadratic temporal rise is due to the increasing cross section, $\pi R_{CB}^2 \propto t^2$ of the fast-expanding CB when it is still opaque to radiation. When it becomes transparent its effective cross section for ICS becomes a constant: the Thomson cross section times the number of electrons in the CB. That and the density $n_g$ of the ambient photons—which for a distance $r = \gamma \delta c t/(1+z) > R_g$ (the radius of the glory) decreases like $n_g(r) \propto 1/r^2 \propto 1/t^2$– contributes a factor $t^{-2}$ to the late temporal decline.

If CBs are launched along the axis of the glory of a torus-like pulsar wind nebula, or of an accretion disk with a radius $R_g$, the glory photons at a distance $r$ from the center intercept the CB at an angle $\arccos[r/(r^2 + R_g^2)^{1/2}]$ resulting in a $t$-dependent peak energy

$$\mathcal{E}_p(t) \approx \mathcal{E}_p(0)[1 - t/(t^2 + \tau^2)^{1/2}] \tag{7}$$

with $\tau = R_g(1+z)/\gamma \delta c$, valid also at late times for an approximately spherical glory.

For LGRBs with $\tau \gg \Delta$, Equations (6) and (7) yield half maximum values at $t \simeq 0.41\Delta$ and $t \simeq 2.41\Delta$, resulting in a full width at half maximum (FWHM) with a value $\simeq 2\Delta$, a rise time from half maximum to peak value RT $\simeq 0.59\Delta$ and a decay time from peak count to half peak, DT $\simeq 1.41\Delta$. Consequently RT/DT $\simeq 0.42$ and RT $\simeq 0.30$ FWHM. The pulse shape given by Equations (6) and (7), with $\Delta = 7.1$ s and $\tau = 70.1$ s, is shown in Figure 6 for the particularly well measured single-pulse of GRB930612 [80].

In most LGRBs $\tau \gg \Delta$ and the CB model predicts RT/DT $\approx 0.42$, and RT/FWHM $\approx 0.29$, changing very little with $\tau$ if $\tau \gg \Delta$. Even in the very rare cases where $\tau/\Delta \approx 1$, RT/DT $\approx 0.57$ and RT/FWHM $\approx 0.36$. In Figures 7 and 8 the predicted ratios RT/DT and RT/FWHM for $\Delta < \tau < \infty$ are compared to their best fit values in the 77 resolved pulses of BATSE/CGRO LGRBs reported in [80]. As shown in these figures their best fit values lie well within the narrow area between the predicted CB-model boundaries. The mean observed values, RT/DT $= 0.47 \pm 0.08$ and RT/FWHM $= 0.31 \pm 0.03$ [80], are very close to the CB-model's expected values RT/DT $= 0.44$ and RT/FWHM $= 0.31$ for $\tau = 10\,\Delta$.

In Figure 9 the measured and CB-model pulse shapes of SHB170817A are compared. The best-fit light curve has a maximum at $t = 0.43$ s, half maxima at $t = 0.215$ s and $t = 0.855$ s, RT/DT $= 0.50$ and RT/FWHM $= 0.34$.

### 3.3.2. Pulse Shapes in Fireball Models

In current standard FB models [24,41–52], the GRB prompt pulses are produced by synchrotron radiation from shock-accelerated electrons in collisions between overtaking thin shells ejected by the central engine, or by internal shocks in the ejected conical jet. Only for the fast decline phase of the prompt emission, and only in the limits of very thin shells and fast cooling, falsifiable predictions have been derived. In these limits the fast decline

phase of a pulse was obtained from the relativistic curvature effect [80–88]. It yielded a power law decay $F_\nu(t) \propto (t-t_i)^{-(\beta+2)} \nu^{-\beta}$, where $t_i$ is the beginning time of the decay phase, and $\beta$ is the spectral index of prompt emission.

The observed decay of the SHB170817A pulse, accompanied by a fast spectral softening before the afterglow took over, could be roughly reproduced by adjusting a beginning time of the decay and replacing the constant spectral index of the FB model by the observed time-dependent one [80–86].

No universal shape of the GRB and SHB prompt emission pulses consistent with their observed temporal and spectral behavior has been published. See, e.g., the FB model's predicted shape in Figure 1 of [89].

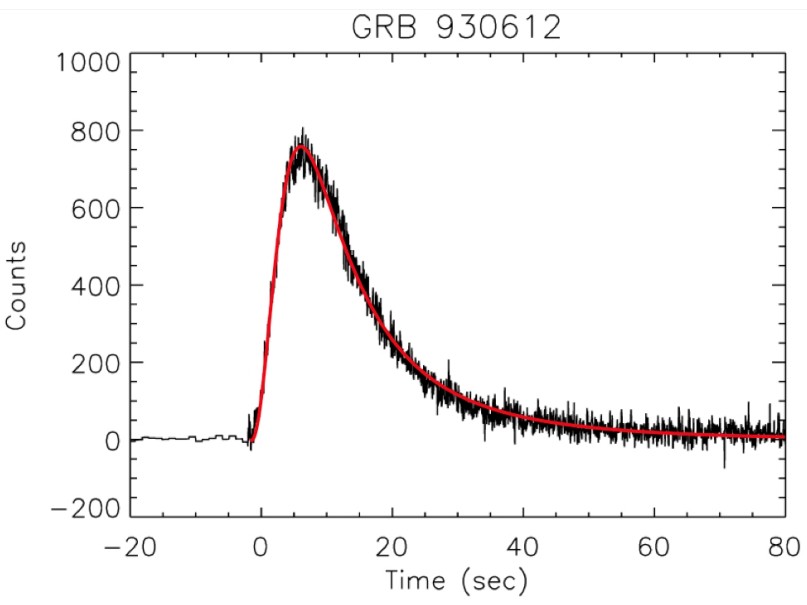

**Figure 6.** The pulse shape of GRB930612 measured with BATSE aboard CGRO and the shape given by Equation (6) for $\Delta = 7.1$ s and $\tau = 70.1$ s, the best CB-model fit to re-bined data.

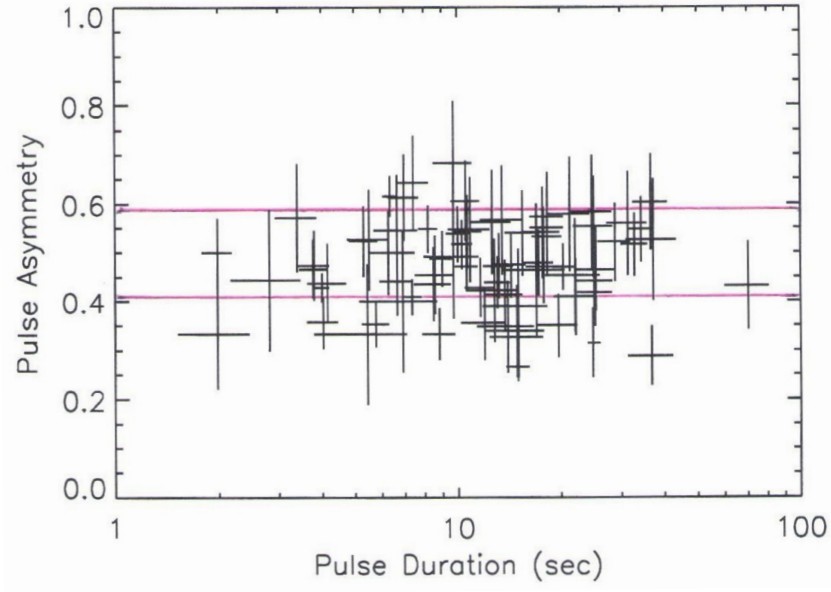

**Figure 7.** Comparison between the observed asymmetry ratio RT/DT as function of pulse duration reported in [80] for a sample of 77 resolved LGRB pulses measured with BATSE aboard CGRO (with a mean value RT/DT = 0.47 ± 0.08), and the CB model predicted asymmetry 0.41 < RT/DT < 0.58 for $\Delta < \tau < \infty$ (solid lines).

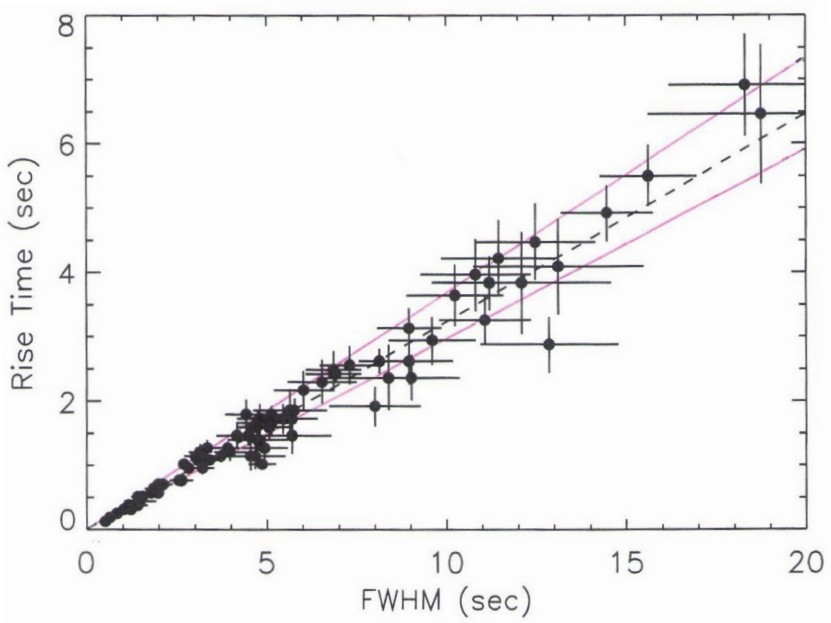

**Figure 8.** Comparison between the rise time RT versus the FWHM reported in [80] for a sample of 77 resolved pulses measured with BATSE aboard CGRO. The dotted line is best fit ratio RT/FWHM = 0.32 and the solid lines are CB model expected boundaries 0.29 < RT/FWHM < 0.36 for LGRBs.

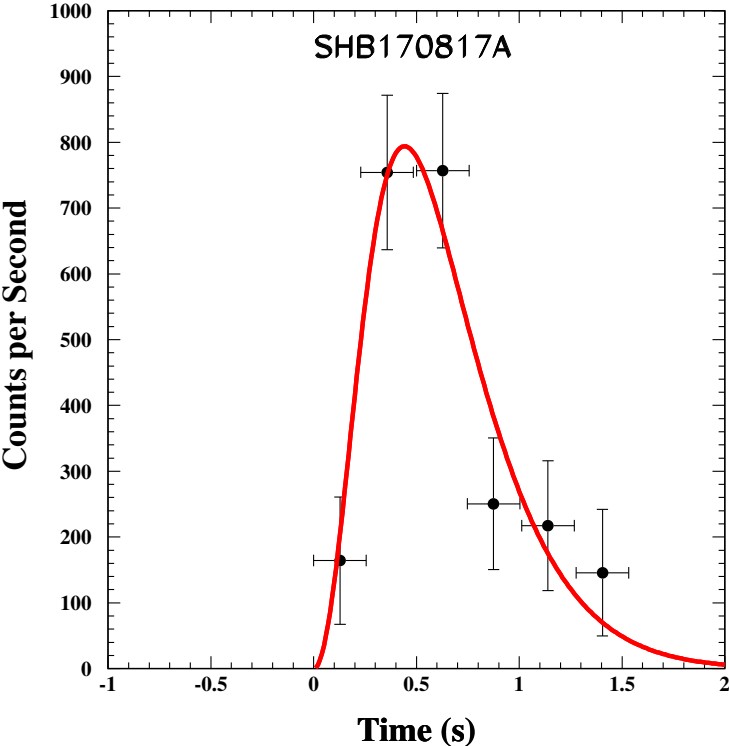

**Figure 9.** The pulse shape of SHB170817A measured with the Fermi-GBM [25–29] and the best fit pulse shape given by Equations (6) and (7) with $\Delta = 0.62$ s and $\tau = 0.57$ s, $\chi^2/\text{dof} = 0.95$.

## 4. The Afterglow of GRBs

In the CB model, the afterglow of SN-associated GRBs (SN-GRBs) is mainly synchrotron radiation (SR) from the relativistic jets of CBs launched in core collapse SNe of type Ic into the dense interstellar medium—such as a molecular cloud—where most SNeIc

take place. The AG of SN-less GRBs and SHBs is dominated by a pulsar wind nebula (PWN) emission powered by the spin down of a newly-born millisecond pulsar [39,40].

The colossal energy of the collimated $\gamma$ rays of a GRB is more than sufficient to fully ionize the interstellar medium (ISM) it travels through. In SN-GRBs, this ionized medium is swept into the CBs and generates within them a turbulent magnetic field. Its magnetic energy density is assumed to be in approximate equipartition with that of the swept in particles, as indicated by simulations [90] and by the equilibrium between the energy densities of galactic cosmic rays and magnetic fields.

A CB's Lorentz factor $\gamma(t)$ decreases as it collides with the ISM. The electrons that enter a CB are Fermi accelerated there and cool by emission of SR, isotropic in the CB's rest frame. In the observer's frame, the radiation is forward-beamed into a cone of opening angle $\theta \sim 1/\gamma(t)$ and is aberrated, Doppler boosted in energy and redshifted in the usual way [12,53–57].

The observed spectral energy density (SED) flux of the *unabsorbed* synchrotron X-rays, $F_\nu(t) = \nu \, dN_\nu/d\nu$, has the form (see, e.g., Equations (28)–(30) in [57,91]),

$$F_\nu \propto n(t)^{(\beta_x+1)/2} \, [\gamma(t)]^{3\,\beta_x-1} \, [\delta(t)]^{\beta_x+3} \, \nu^{-\beta_x} \,, \tag{8}$$

where $n$ is the baryon density of the external medium encountered by the CB at a time $t$ and $\beta_x$ is the spectral index of the emitted X-rays, $E \, dn_x/dE \propto E^{-\beta_x}$.

The CBs are decelerated by the swept-in ionized material. Energy-momentum conservation for such a plastic collision[1] between a CB of baryon number $N_B$, radius $R$ and initial Lorentz factor $\gamma_0 \gg 1$, propagating in a constant-density ISM, decelerates according to [79]:

$$\gamma(t) = \frac{\gamma_0}{\left[ \sqrt{(1 + \theta^2 \, \gamma_0^2)^2 + t/t_d} - \theta^2 \, \gamma_0^2 \right]^{1/2}} \,, \tag{9}$$

where $t$ is the time in the observer frame since the beginning of the AG emission by a CB, and $t_d$ is its deceleration time-scale

$$t_d = (1+z) \, N_B / 8 \, c \, n \, \pi \, R^2 \, \gamma_0^3. \tag{10}$$

In the case of SN-less LGRBs the AGs—to be discussed in detail below– are compatible with the radiation emitted by the pulsar's wind nebula, powered by the rotational energy loss of the newly born quark star through magnetic dipole radiation, relativistic wind and high energy charged-particle emission along open magnetic field lines [40].

### 4.1. "Canonical" Behavior of the AG of LGRBs (Test 4)

In the CB model the prompt $\gamma$-ray emission was predicted to end with the exponential temporal decay and fast spectral softening of Equation (6), subsequently taken over by a *canonical* X-ray AG, i.e., an initial shallow decay phase (the *plateau*) that breaks smoothly into a power-law decline. The shallow decay phase lasts until the time when the initial rest mass of the CB is approximately doubled by the swept-in relativistic mass. In Figures 10 and 11 this behaviour of the X-ray AG of two LGRBs is shown to agree with the CB model prediction (see, e.g., Figures 27–33 in [57,91]). This AG-shape prediction was made long before it was first observed with the Swift X-ray Telescope in the AGs of GRB050315 [92] and GRB050319 [93] and subsequently categorized as "canonical".

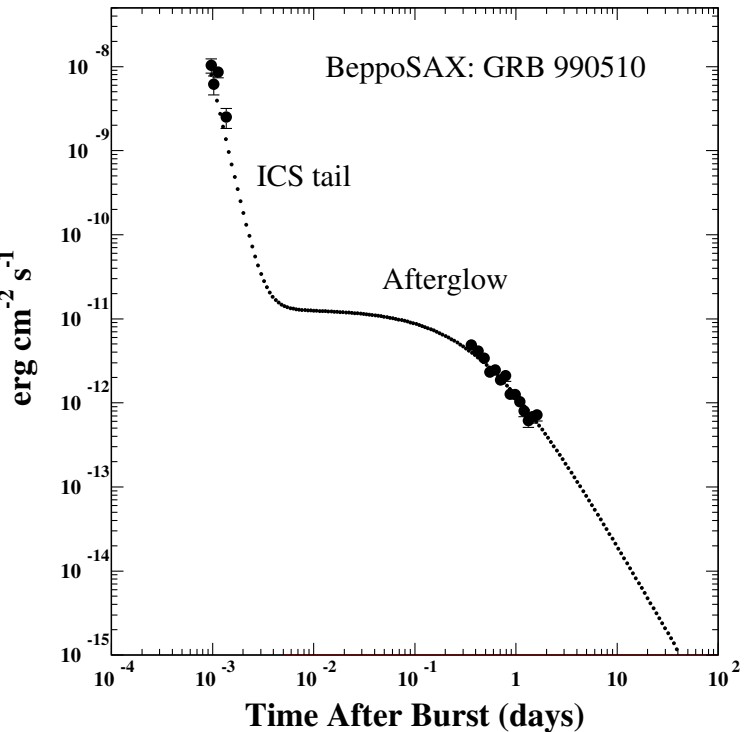

**Figure 10.** The X-ray afterglow of GRB 990510 measured with the telescopes aboard the BeppoSAX satellite compared to the canonical X-ray afterglow predicted by the CB model [57,91] of SN-GRBs.

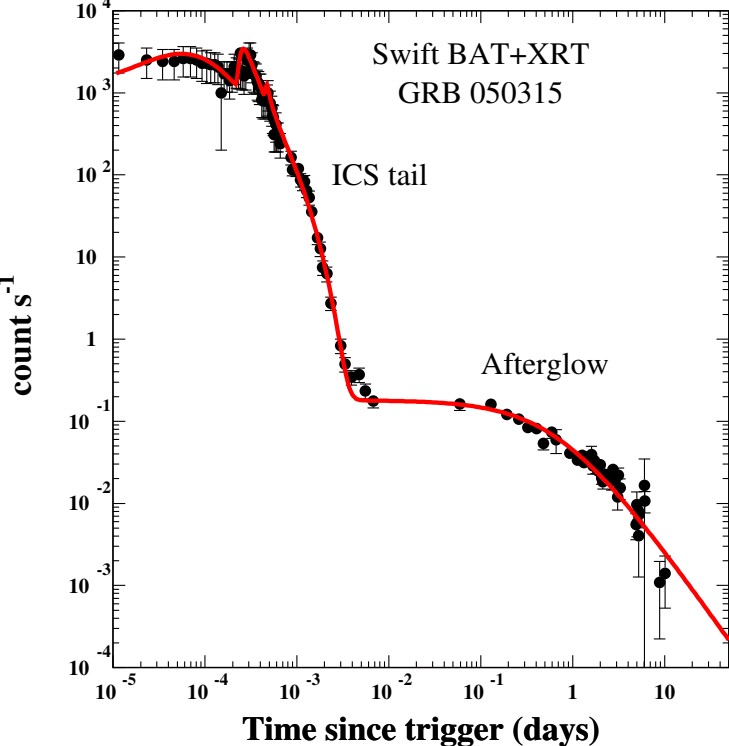

**Figure 11.** The X-ray afterglow of GRB050315 measured with the telescopes aboard Swift compared to its best fit canonical X-ray afterglow predicted by the CB model [57,91] for SN-GRBs.

### 4.2. Break Time Correlations (Test 5)

In the CB model, the Lorentz factor $\gamma(t)$ of a CB, decelerating as per Equation (9), changes slowly until $t_b$, a break time approximately given by [79]:

$$t_b \approx (1+\gamma_0^2\theta^2)^2 t_d, \tag{11}$$

where $t_d$ as in Equation (10). This slow change is responsible for the plateau phase of SN-GRBs.

The dependence of $E_p$ and $E_{iso}$ on $\gamma_0$ and $\delta_0$ can be used to obtain from Equation (10) the correlation [94]

$$t_b/(1+z) \propto [(1+z)\,E_p\,E_{iso}]^{-1/2}, \tag{12}$$

The observed break time of the X-ray afterglow of LGRBs measured with the Swift satellite XRT (X Ray Telescope) [95,96] for LGRBs with known $z$, $E_p$ and $E_{iso}$, is compared in Figure 12 to that predicted by Equation (11), with satisfactory results.

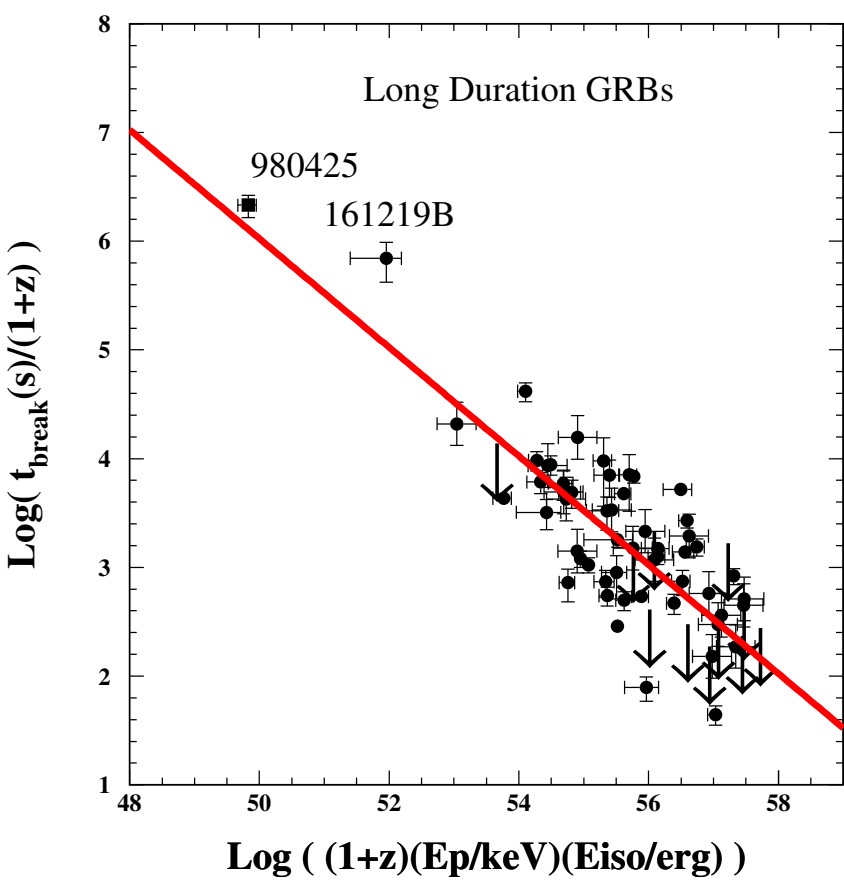

**Figure 12.** The break time $t_b/(1+z)$ of the X-ray AG of SN-GRBs measured with the Swift XRT [95,96], as a function of $[(1+z)\,E_p\,E_{iso}]$. The line is the CB model correlation of Equation (6), expected in SN-GRBs. SN-Less GRBs, to be discussed in Section 6.2, are not included.

### 4.3. Post-Break Closure Relations (Test 6)

Well after the break, Equation (9) yields $\delta(t) \approx 2\gamma(t) \propto t^{-1/4}$, which, when substituted in Equation (8), results in the late-time behavior

$$F_\nu(t \gg t_b) \propto t^{-\alpha_\nu} E^{-\beta_\nu}, \tag{13}$$

and the *closure* relation

$$\alpha_\nu = \beta_\nu + 1/2. \tag{14}$$

This relation, exact for a constant ISM density, is well satisfied by the X-ray AG of SN-GRBs [94]—shown in Figure 13 for the canonical case of GRB060729 [95,96]—as long as the CBs move in an ISM of roughly constant density. Indeed, this long followed up and well X-ray AG is best fit with a temporal index $\alpha_x = 1.46 \pm 0.03$, which agrees well with the predicted $\alpha_x = \beta_x + 1/2 = 1.49 \pm 0.07$ for an observed [95,96] $\beta_x = 0.99 \pm 0.07$.

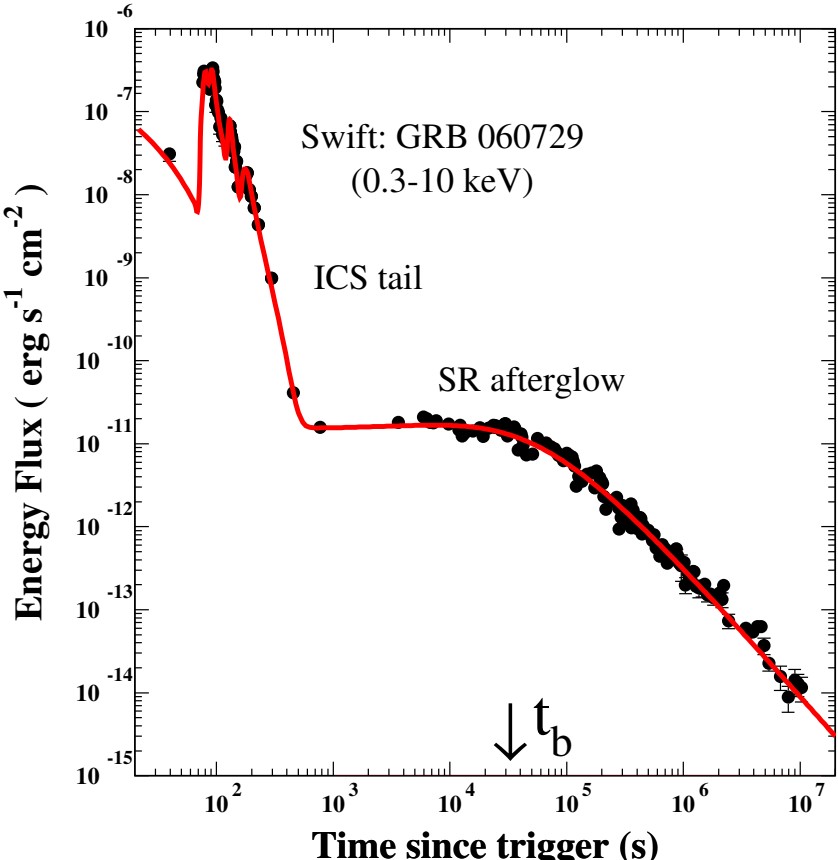

**Figure 13.** The canonical light curve of the X-ray AG of the SN-GRB 060729 measured with Swift XRT [95,96], and its best fit CB-model AG [12,53–57], given by Equation (8). The data also satisfy the CB model prediction $\alpha_x = \beta_x + 1/2$.

The most accurate test of the CB model prediction of Equation (14) for a single SN-GRB was provided by the measurements of the X-ray AG of GRB 130427A, the most intense GRB ever detected by Swift and followed with the Swift XRT and X-ray telescopes aboard XMM Newton and CXO up to a record time of 83 Ms after burst [97]. The measured light-curve has a single power-law decline with $\alpha_x = 1.309 \pm 0.007$ in the time interval 47 ks to 83 Ms. The best single power-law fit to the combined measurements of the X-ray light-curve of GRB 130427A with the Swift-XRT [21], XMM Newton, CXO [97], and MAXI [98], shown in Figure 14 yields $\alpha_x = 1.294 \pm 0.03$. The CB model prediction, as given by Equation (14) for the measured spectral index $\beta_x = 0.79 \pm 0.03$ [97], is $\alpha_x = 1.29 \pm 0.03$, in remarkable agreement with its best fit value.

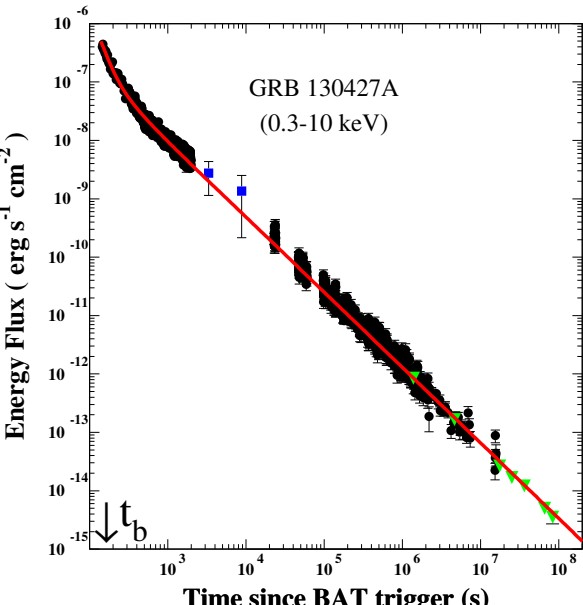

**Figure 14.** The X-ray light-curve of the intense GRB 130427A, measured with Swift XRT [95,96] (circles) and with XMM Newton and Chandra [97] (triangles) up to 83 Ms after burst, and its CB model best-fit with a start time and an early break hidden under the prompt emission phase. The two MAXI data points [98] (squares) at $t = 3257$ s and $t = 8821$ s are also shown. The best-fit power-law decline has an index $\alpha_x = 1.29$. The temporal decay index predicted by the CB model, Equation (14), for the measured spectral index [97] $\beta_x = 0.79 \pm 0.03$ is $\alpha_x = 1.29 \pm 0.03$.

No doubt, the assumptions of a constant density circumburst medium is an over simplification: SN-LGRBs are produced by SN explosions of type Ic, which generally take place in super-bubble remnants of star formation. These environments may have a bumpy density, which deviates significantly from an assumed constant-density ISM. This explains, in the CB model, the observed deviations from the predicted smooth light-curves [99].

In a constant-density ISM, the late-time distance $x$ of a CB from its launch site is:

$$x(t \gg t_b) = \frac{2\,c \int^t \gamma\,\delta\,dt}{1+z} \approx \frac{4\,c\,\gamma_0^2\,\sqrt{t_b\,t}}{1+z}\,. \tag{15}$$

This distance can exceed the size of the super-bubble and even the scale-height of the disk of a GRB's host galaxy. In such cases, the transition of a CB from the super-bubble into the Galactic ISM or into the Galactic halo in face-on disk galaxies, will bend the late-time single power-law decline into a more rapid decline, depending on the density profile above the disk. For instance, when the CB exits the disk into the halo, its Lorentz and Doppler factors tend to constant values while, given Equation (8), its AG decays like:

$$F_\nu(t) \propto [n(r)]^{(1+\beta_\nu)/2}, \tag{16}$$

with $r \propto t$. This behavior may have been observed with Swift's XRT [95,96] in several GRBs, such as 080319B and 110918A, at $t > 3 \times 10^6$ s and by CXO in GRB 060729 at $t > 3 \times 10^7$ s [100].

### 4.4. Missing Breaks (Test 7)

The CB model's Equation (8) implies a single power-law for the temporal decline of the light curve of an AG well beyond its break time $t_b$, as long as the CB moves in a constant density interstellar medium. Consequently very energetic LGRBs, i.e. those with a large product $(1 + z)E_p E_{iso}$, may have a break time $t_b$ smaller than the time at which the AG takes over the prompt emission, or before the AG observations began [101]. In such cases,

the observed light curve has a single power-law behavior with a temporal decay index $\alpha_\nu = \beta_\nu + 1/2$ and the break is missing.

The first *missing break*, shown in Figure 15, was observed in GRB061007 [102] with Swift's XRT. The $\alpha_x$ values of the most energetic Swift XRT LGRBs with known redshift are plotted in Figure 16 along with their $\beta_x + 1/2$ values. The best fit linear relation $\alpha_x = a(\beta_x + 1/2)$ is also plotted, which yields $a = 1.007$, in agreement with $a = 1$, predicted by the CB model.

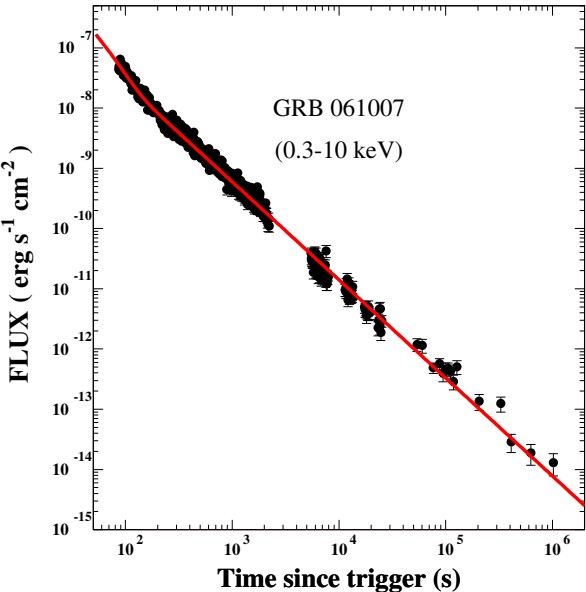

**Figure 15.** The single power-law best fit to the afterglow of GRB061007 with a "missing jet break" measured with Swift XRT [102]. The best fit temporal index $\alpha_x = 1.65 \pm 0.01$ satisfies the CB model prediction $\alpha_x = \beta_x + 1/2 = 1.60 \pm 0.11$.

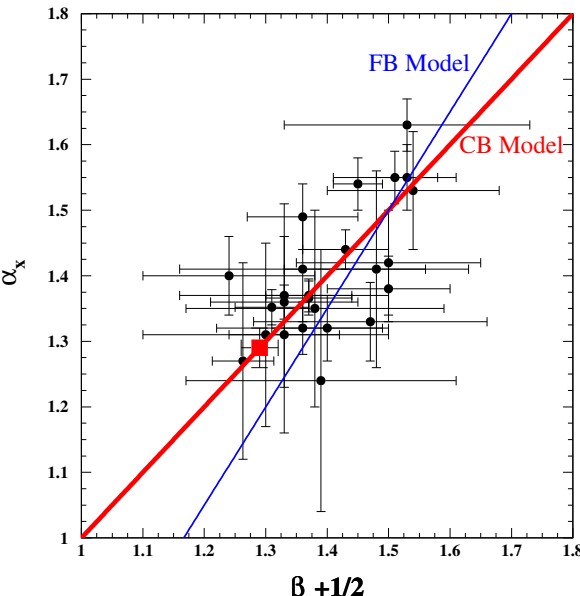

**Figure 16.** The values of the post-break temporal index $\alpha_x$ as a function of the spectral index $\beta_x + 1/2$ for the 28 most intense GRBs with known redshift [101] obtained from the follow-up measurements of their 0.3–10 keV X-ray AG with Swift's XRT [95,96]. The square indicates the result for GRB 130427A. The thick line is the CB model prediction Equation (14), the thin one is the FB's model. The models have similar $\chi^2/\text{dof}$ but 3/4 of the points lie above the FB-model prediction.

## 5. GRB Afterglows in Fireball Models

As illustrated in Figure 2, the prompt GRB emission is emitted by shells colliding with each other while moving in the same direction, unlike in a particle collider. The AG is produced by the ensemble of the shells stopping against the ISM, like in a beam dump. The center-of mass energy in this last process is much bigger than in the first mentioned one. This implies that the prompt emission's energy ought to be orders of magnitude smaller that the AG's one, exactly contrary to observation.

During the first two years after the discovery of afterglows, the observed light curves were claimed in FB models to be well fitted by a single power-law [103], as predicted for spherical fireballs [15,41]. Shortly after, when the observations clearly showed a smoothly broken power-law, the spherical $e^+e^-\gamma$ fireball was replaced first with conical flows or thin conical shells of $e^+e^-\gamma$ plasma, later to be *baryon-loaded* and eventually to consist in an ordinary-matter plasma. These flows were called *collimated fireballs*, retaining the fire *ball* denomination. The collimated fireballs could neither explain nor correctly reproduce the observed behavior of the AG of SN-GRBs: they failed tests 4 to 7.

### 5.1. The Canonical AG Shape (Test 4)

In the FB models, a plateau phase in the AG of GRBs was not expected [24,41–52]. An additional power supply by a newly born millisecond pulsar was later adopted as the origin of such a phase [104].

### 5.2. Break-Time Correlations (Test 5)

In the standard FB models, the opening angle of the conical jet satisfies $\theta_j \gg 1/\gamma_0$. Because of relativistic beaming, only a fraction $\sim 1/\gamma^2 \theta_j^2$ of the front surface of the jet is initially visible to a distant near axis observer. This fraction increases with time like $[\gamma(t)]^{-2}$, due to the deceleration of the jet in the ISM, until the entire front surface of the jet becomes visible, i.e. until $t \approx t_b$, with $\gamma(t_b) = 1/\theta_j$. If the total $\gamma$-ray energy $E_\gamma$ is assumed to be a constant fraction $\eta$ of the kinetic energy $E_k$ of the jet, which decelerates in an ISM of constant density $n_b$ via a *plastic collision*, it follows that [105,106]:

$$t_b/(1+z) \approx \frac{1}{16\,c} \left[ \frac{3\,E_{iso}}{\eta\,\pi\,n_b\,m_p\,c^2} \right]^{1/3} [\theta_j]^{8/3}. \tag{17}$$

Though in the FB model the AG is SR by the shocked ISM—through *elastic scattering* of the ISM particles in front of the jet, not by a *plastic collision*, see Table 2—Equation (17) has been widely used, with no rationale, to estimate $\theta_j$.

**Table 2.** The late time t-dependence of the bulk motion Lorentz factor of highly relativistic conical jets which decelerate by collision with the surrounding medium.

| Collision: | Plastic | Plastic | Elastic | Elastic |
|---|---|---|---|---|
| Density: | ISM | Wind | ISM | Wind |
| $\gamma(t) \propto$ | $t^{-3/8}$ | $t^{-1/4}$ | $t^{-3/7}$ | $t^{-1/3}$ |

If $E_\gamma \approx \eta\,E_k \approx E_{iso}\theta_j^2/4$ is a *standard candle*, as argued in [107], $E_{iso}\,\theta_j^2$ is roughly constant (the "Frail relation") and

$$t_b/(1+z) \propto [E_{iso}]^{-1}. \tag{18}$$

The same $[t_b, E_{iso}]$ correlation is obtained for the deceleration of a conical jet in a wind-like circumburst density [108,109]. Equation (18) is inconsistent with $t_b/(1+z) \propto E_{iso}^{-0.69\pm0.06}$, the best fit to the data, but it is consistent with the correlation $t_b/(1+z) \propto E_{iso}^{-3/4}$ expected in the CB model [94].

### 5.3. Closure Relations (Test 6)

In the conical fireball model the increase of the visible area of the jet until the break time results in an achromatic break in the AG. If the spectral energy density flux is parametrized as $F_\nu(t) \propto t^{-\alpha}\nu^{-\beta}$, the predicted achromatic change in $\alpha$ across the break satisfies $\Delta(\alpha) = \alpha(t > t_b) - \alpha(t < t_b) = 3/4$ for a constant ISM density. For a wind-like density, $\Delta(\alpha) = 1/2$. None of these closure relations is satisfied by most GRB breaks, e.g., the ones of Figures 10, 11 and 13–15.

Liang, et al. [110] analyzed the AG of 179 GRBs detected by Swift between January 2005 and January 2007 and the optical AG of 57 pre-Swift GRBs. They did not find any afterglow with a break satisfying tests 5 or 6 of the FB model.

### 5.4. Missing Breaks (Test 7)

It was hypothesized that the missing break in the X-ray AG of several GRBs with long follow up measurements took place after the observations ended [111]. But Equation (17) implies that late-time breaks are present only in GRBs with a small $E_{iso}$. This contradicts the fact that missing breaks in GRBs extending to late times are limited to events with very large, rather than small, $E_{iso}$. This is demonstrated in Figures 14 and 15 by the unbroken power-law X-ray AGs of GRBs 061007 and 130427A, for which $E_{iso} = 1.0 \times 10^{54}$ erg [102] and $E_{iso} = 8.5 \times 10^{53}$ erg [97] respectively. These AGs satisfy the CB model post-break closure relation given by Equation (12).

## 6. Further Afterglow Tests

### 6.1. Chromatic Jet Breaks (Test 8)

In the CB model the jet deceleration break in the afterglow of jetted SN-GRBs is dynamic in origin and usually chromatic [12,53–57], while in the FB model they are basically geometrical in origin and are therefore predicted to be dominantly achromatic [24,41–52], in conflict with observations.

### 6.2. The Universal Afterglow of SN-Less GRBs (Test 9)

Figure 17, adapted from [112], shows the X-ray AG of GRB 990510 measured with BeppoSAX. It could not be fit by the single power-law predicted by spherical FB models (e.g., [112]). But it could be fit well by an achromatic "smoothly broken power law" parametrization [112], as shown in Figure 18. That—and the observed optical and X-ray AGs of a couple of other GRBs, which could be fit by such a parametrization—is what led to the replacement of the spherical $e^+e^-\gamma$ fireball by a conical one, later replaced by a conical jet of ordinary matter which became the current FB model of GRBs.

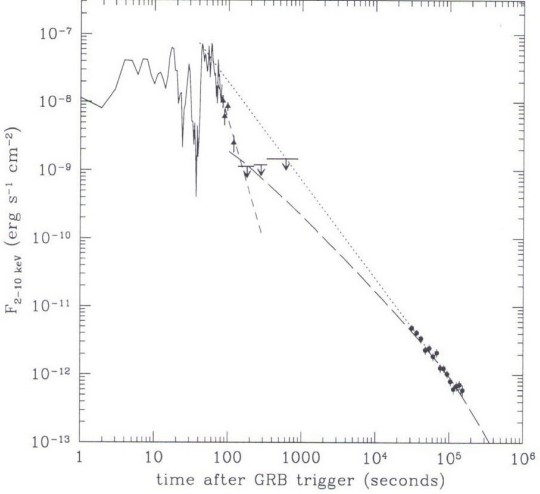

**Figure 17.** The X-ray light curve of GRB 990510 and its AG, measured with BeppoSAX together with a single power-law fit and a smoothly broken single power-law fit to its X-ray AG [112].

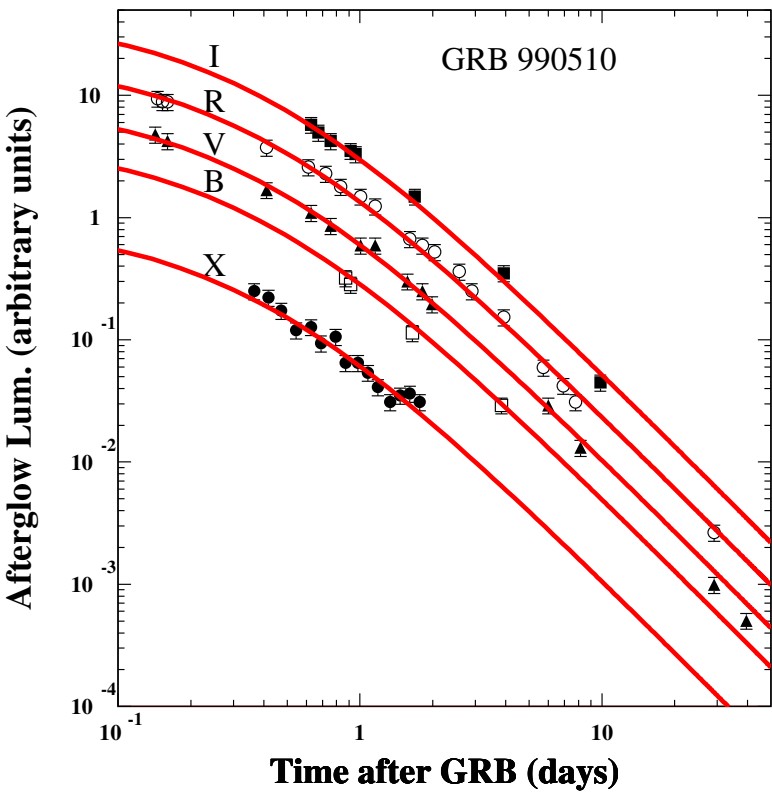

**Figure 18.** Comparison of Equation (19), the predicted temporal behavior of the light curves of the X-ray and optical AGs of GRB990510, with the observations. The X-ray data at 5 keV (filled circles) are from [112]. The data in the bands I (filled squares), R (empty circles), V (filled triangles) and B (open squares) are those compiled in [112] from [113–116]. The flux normalization is arbitrary.

The AG of GRB 990510 and others which were fit by smoothly broken power laws are not conclusive evidence of conical jets. In fact, an isotropic radiation from a pulsar wind nebula (PWN), powered by a newly born millisecond pulsar, has an expected luminosity [117] satisfying

$$L(t, t_b)/L(t = 0) = (1 + t/t_b)^{-2},$$ (19)

where $t_b = P(0)/2\dot{P}(0)$, and $P(0)$ and $\dot{P}(0)$ are the pulsar's initial period and its time derivative. This is shown in Figure 18 for GRB 990510. The *universal behaviour* [40] of Equation (19) describes well the X-ray and optical AG light curves of GRB 990510, as shown in Figures 18 and 19 and of the AG of all the SN-less GRBs and SHBs with a well sampled AG during the first few days after burst. This is demonstrated in Figure 20 for the X-ray AG of twelve SN-less GRBs, and in Figure 21 for the twelve SHBs [34] from the Swift XRT light curve repository [95,96], that were well sampled in the mentioned period.

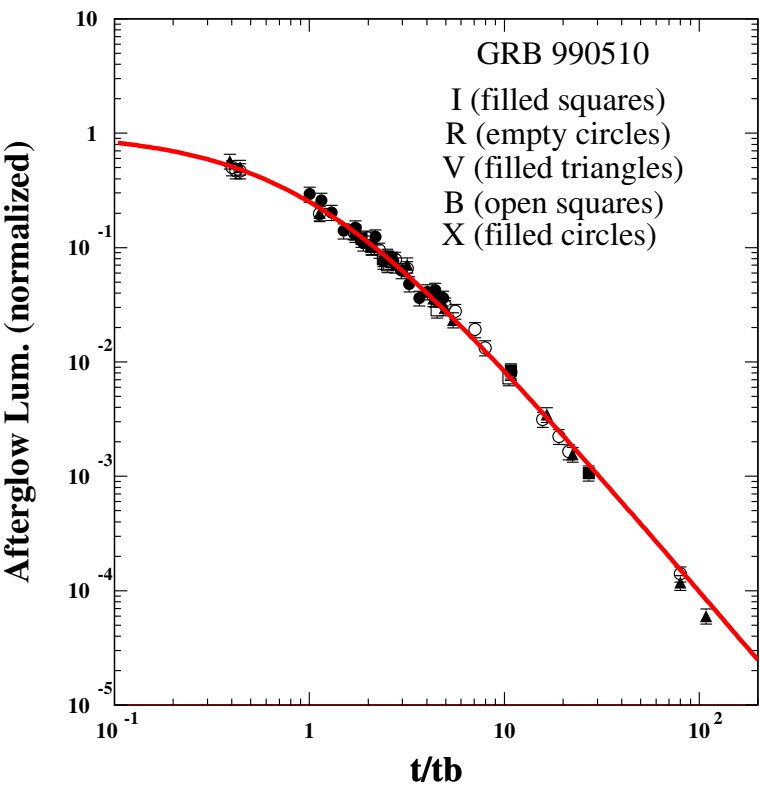

**Figure 19.** Comparison between the normalized light curves of the X-ray and optical AGs of GRB990510 and their universal shape: Equation (19). The data are the same as in Figure 18.

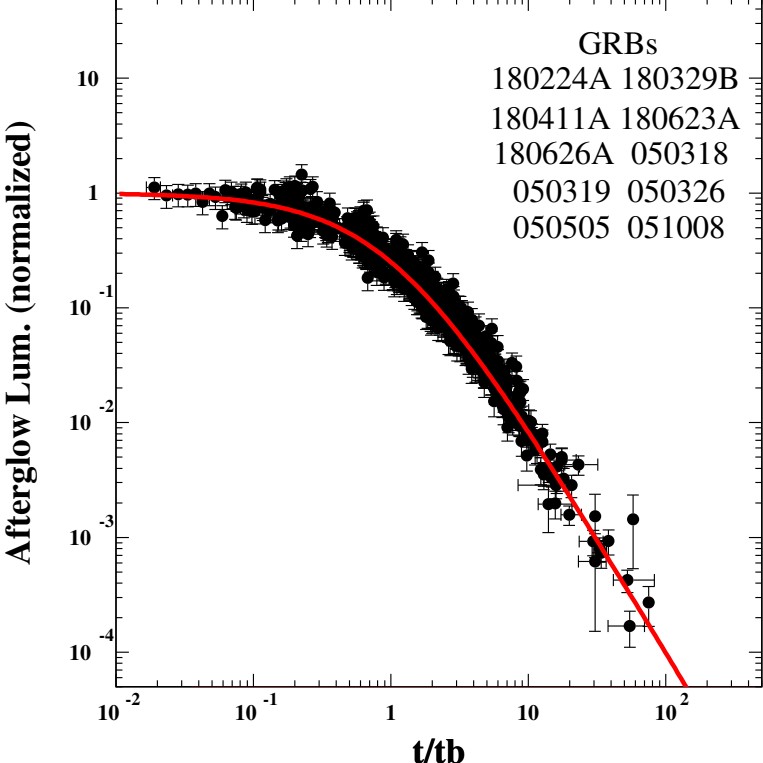

**Figure 20.** Comparison between the normalized light curve of the X-ray afterglow measured with Swift XRT [95,96] of 12 SN-less GRBs with a well sampled AG in the first couple of days after burst and their predicted universal behavior, Equation (19).

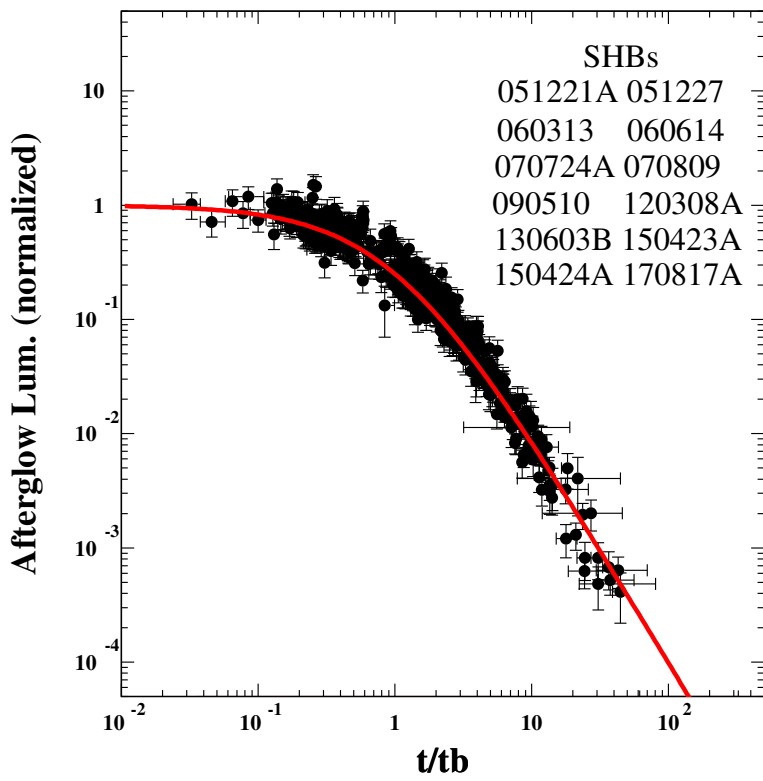

**Figure 21.** Comparison between the normalized light curve of the X-ray AG of 11 SHBs with a well sampled AG measured with Swift's XRT [95,96] during the first couple of days after burst and the predicted universal behavior of Equation (19). The bolometric light curve of SHB170817A [118] is included.

## 7. The Progenitors of GRBs

The possibility that GRBs be produced in SN explosions was suggested very long ago [8,119–121]. This hypothesis was incorporated in the CB model as the daring bet that there ought to be a 1:1 correspondence between LGRBs and Type Ic SNe, for a long time a tenable opinion [12,53–57]. The first direct evidence for an SN-GRB association was provided by the discovery [21]—within the Beppo-SAX error circle around the position of GRB980425 [122,123]—of SN1998bw, at $z = 0.0085$. The SN light curve indicated that the time of the explosion was within $-2$ to $+7$ days of the GRB [21,124]. A physical association between GRB980425 and SN1998bw was consistent with the GRB being entirely normal, produced by a relativistic jet in an SN explosion and viewed at a relatively large angle [12,125]. Our explicit CB model analysis was entirely consistent with this, as can also be seen in Figure 4.

During the first few years after GRB980425, the optical afterglow of several relatively nearby ($z < 0.5$) GRBs provided clear evidence for a GRB-SNIc association, independent of GRB luminosity. The most convincing evidence was provided by the fairly nearby ($z = 0.1685$) GRB030329 [126], one of the brightest GRBs detected by that time. The CB-model's fit to its early AG data was extrapolated, with the addition of a superimposed SN1998bw light curve (transported in distance) to predict the exact date when an SN would be discovered as a bump in the AG and a dramatic change in the spectrum [127]. SN2003dh duly obeyed the CB model's forecast. Admittedly, there was an element of timing luck in all this: SNLess GRBs were discovered later and they proved wrong the original hypothetical 1:1 SN/GRB association.

An early standard view of the SN/GRB association was that of Woosley [58,59], who argued that GRBs were not be produced by SNe, but by "failed SNe", or "collapsars", direct collapses of massive stars into black holes without an associated supernova. Concerning SN1998bw and GRB980425, the FB model defenders concluded that this pair

represented a new subclass of rare events [21,128–130]. These would be associated with "hyper-novae" [6,62,124], super-energetic explosions with kinetic energy exceeding $10^{52}$ erg, as was inferred for SN1998bw from its high expansion velocity and luminosity [131] and from the very strong radio emission from its direction [132,133].

## 8. The Progenitors of SHBs

After the discovery of the gravitational wave GW170817 and its electromagnetic sibling SHB170817A, a consensus was reached that SHBs originate in the merger of neutron star pairs. Two days before the discovery date, a paper appeared on arXiv [134], not only reiterating that earlier view, but predicting that an SHB found in this "multi-messenger" way would be seen far of axis. This fact, also currently generally accepted, is based on the greater red-shift reach of GRB or SHB observations relative to the GW ones. In the CB model, within the volume reach of GW observations, it would be most unlikely for a SHB to point close to the observer.

## 9. Further Tests

### 9.1. Redshift Distribution of LGRBs (Test 10)

In the CB model long duration GRBs belong to two classes, SN-GRBs produced by Type Ic SNe and SN-less GRBs presumably produced in phase transitions of neutron stars to quark stars following mass accretion in high mass X-ray binaries [12,34]. In both cases, the progenitors involve a short-lived high mass star. Hence the production rate of these GRBs is proportional to the star formation rate [135,136] modified by beaming [137].

The CB-model's beaming fraction, $f_b(z)$, depends on the standard $\Lambda$CDM cosmological luminosity distance, $f_b(z) = N/\sqrt{D_L(z)}$. The proportionality factor $N$ is a function of the minimal luminosity for GRB detection. It can be estimated by use of the properties of XRF060218 at a redshift $z_{min} = 0.0331$ [138] with its record lowest prompt-emission peak energy for a Swift GRB, $E_p^{min} = 4.5$ keV. The result is $N \simeq \sqrt{D_L(z_{min})}\,\epsilon_g/E_p^{min}$, with $\epsilon_g = 3$ eV the estimated peak spectral energy of the glory surrounding the progenitor (a Wolf–Rayet star). In Figure 22 this prediction is tested by a comparison with the data.

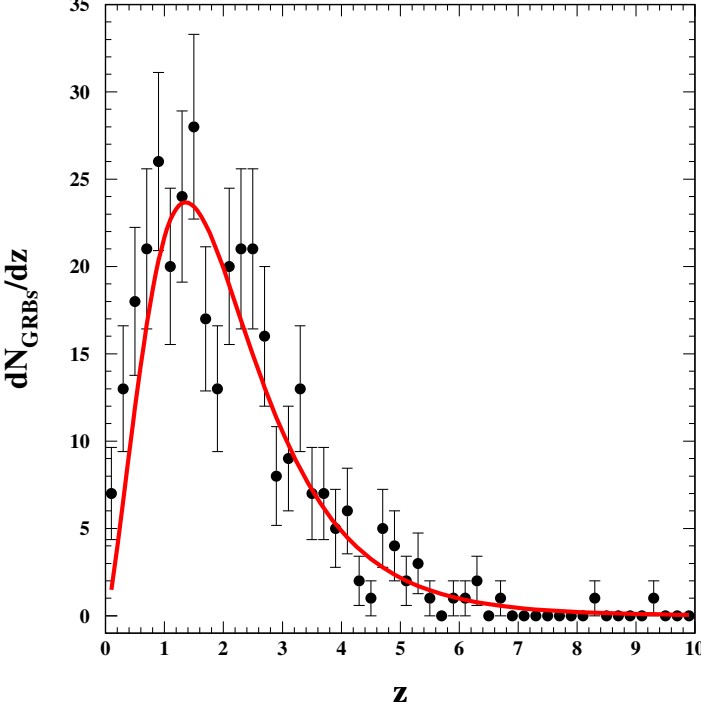

**Figure 22.** Comparison between the redshift distribution of 356 long GRBs with known redshift observed before June 2018 and their CB-model's expected distribution if the production rate of GRBs is proportional to the SFR; ($\chi^2/\text{dof} = 37.57/49 = 0.77$).

In the FB models the jet's aperture obeys $\alpha_j \gg 1/\gamma_0$ so that the rate of GRBs would be expected to be proportional to the star formation rate (SFR) up to very large redshifts beyond those accessible to optical measurements, in contradiction with the data. Unlike the SFR in a comoving unit volume, which first increases with redshift [137], the observed rate of LGRBs decreases with increasing $z$ in the range $z \lesssim 0.1$, even after correcting for detector flux threshold [139]. At larger $z$, it increases faster than the SFR [140–147]. The discrepancy at small $z$ was interpreted as evidence that ordinary LGRBs and low-luminosity LGRBs and XRFs belong to physically distinct classes [139,148–157]. The discrepancy at $z \gg 1$ was claimed to be due to a different evolution [158–160]. In Figure 23 the cumulative distribution, $N(<z)$, of GRBs in CB and FB models are shown. In the latter model, with and without the evolution of LGRBs relative to the SFR assumed in [158–160]. As can be seen in the figure, the data does not support the FB model's assumptions.

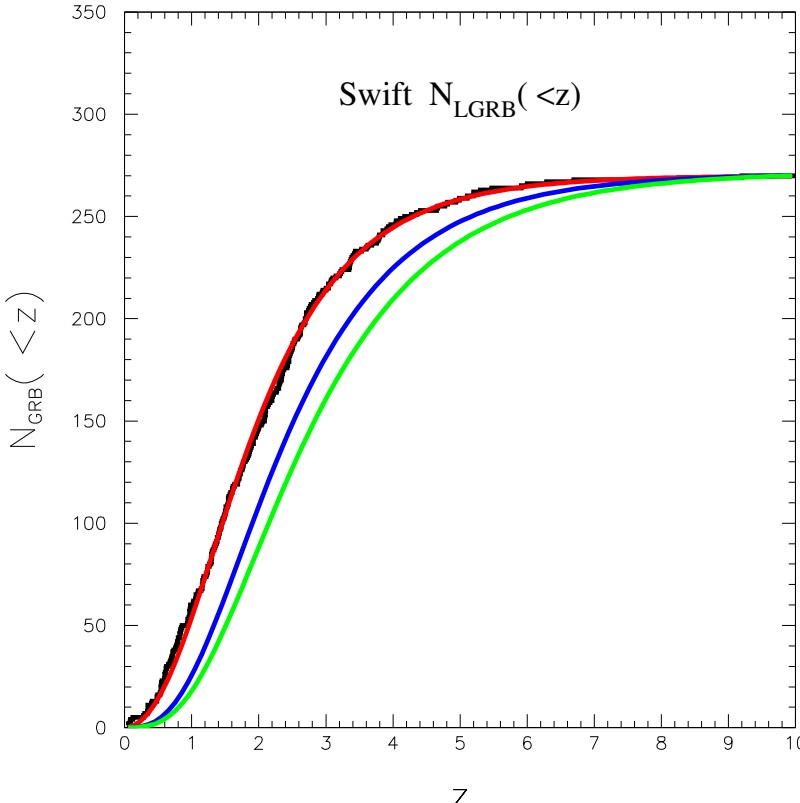

**Figure 23.** Comparison between the cumulative distribution function, $N(<z)$, of the 262 LGRBs with known redshift (histogram) detected by Swift before 2014 and the $N(<z)$ expected in the CB model (left curve) for long GRBs, whose rate is assumed to trace the SFR. The distributions expected in FB models are also shown, with no evolution (rightmost curve) and with it (middle curve) [158–160].

### 9.2. Low Luminosity GRBs (Test 11)

In the CB model the observed properties of GRBs depend strongly on their viewing angle relative to the CBs' direction of motion. Ordinary (O) GRBs are viewed from angles $\theta \sim 1/\gamma_0$, so that their Doppler factor is $\delta_0 \sim \gamma_0$. In the CB model, low luminosity (LL) GRBs are ordinary GRBs with similar intrinsic properties viewed from far off-axis, i.e., $\delta_0\gamma_0 = 1/(1 - \beta_0\cos\theta) \simeq 2/\theta^2$. Consequently, in the rough approximation that GRBs are standard candles, ordinary GRBs and LL GRBs satisfy the relations:

$$E_{iso}(\mathrm{LL\,GRB}) \simeq E_{iso}(\mathrm{O\,GRB})/[\gamma_0^2\,\theta^2]^3 \qquad (20)$$

$$L_p(\mathrm{LL\,GRB}) \simeq L_p(\mathrm{O\,GRB})/[\gamma_0^2\,\theta^2]^4, \qquad (21)$$

between their isotropic-equivalent energy $E_{iso} \propto \gamma_0 \delta_0^3$ and their peak luminosity $L_p \propto \gamma_0^2 \delta_0^4 / (1+z)^2$ [12,53–57]. The best fit CB model light curve to the X-ray AG of GRB980425 [132,161], shown in Figure 24, resulted in $\gamma\theta \approx 8.7$. The mean $E_{iso}$ of ordinary GRBs is $\langle E_{iso}(\mathrm{O\,GRB})\rangle \simeq 7 \times 10^{52}$ erg. Thus, Equation (20) yields $E_{iso}(\mathrm{GRB980425}) \simeq 1.84 \times 10^{-5} \langle E_{iso}(\mathrm{O\,GRB})\rangle \approx 1.3 \times 10^{48}$ erg, agreeing with the observed value [162] $E_{iso}(\mathrm{GRB980425}) \simeq (1.0 \pm 0.2) \times 10^{48}$ erg.

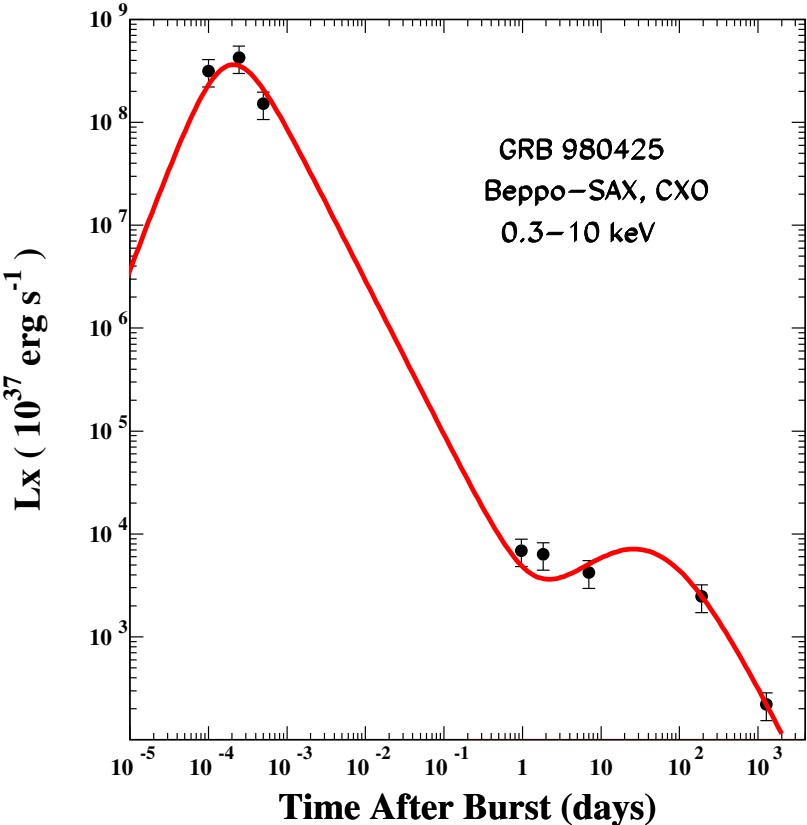

**Figure 24.** The 0.3–10 keV X-ray light-curve of GRB980425 measured by Beppo-SAX [161] (first 7 points). The last point at 1281 days is due to the source S1b resolved by CXO [132,133]. The line is the CB model best fit light-curve to the prompt emission pulse and the AG of GRB980425, yielding $\gamma\theta \approx 8.7$.

Equations (20) and (21), as well as the correlations between other properties of LL GRBs (e.g., the ones tested in Figures 4 and 12) establish that they are ordinary GRBs viewed from far off axis. So does the fact that O and LL GRBs have the same proportionality factor in the relation between their birth rates and the SFR, as in [137] and Figure 23.

One of the best evidences for low-luminosity and ordinary SN-GRBs belonging to the same class is that they are both produced by very similar SNeIc [35,36], akin to SN1998bw. For instance, SN2013cq and its GRB130427A at $z = 0.34$, with a record high GRB fluence and $E_{iso} \sim 10^{54}$ erg [35,36], was very similar to SN1998bw, which produced the LL GRB980425 with a record low $E_{iso} \sim 10^{48}$ erg [162], six orders of magnitude smaller.

In the FB model low luminosity GRBs were claimed to be intrinsically different from ordinary SN-GRBs and to belong to a different class [139,148–157]. A frequent and inescapable conclusion. In this case, given that:

(a) The model could not explain the $\sim 6$ orders of magnitude difference between the $E_{iso}$ of LL SN-GRBs, such as that of GRB980425, and that of very high luminosity SN-GRBs, such as GRB130427A, though they are produced by very similar SNeIc [35,36].

(b) As previously discussed, the FB model cannot describe the GRB distribution as a function of redshift, recall e.g., Figure 23.

The CB model's resolution of these two apparent problems is simple: the interplay between a narrow radiation beam and the observer's direction, occasionally far off-axis. The far off-axis observation of SHB170817A has now been accepted by the FB community, following its measured viewing angle—unexpected in an FB model—relative to the axis of the neutron star binary which produced GW170817 [30–33]—extracted from the behavior of its late time AG [163]—and from the superluminal motion—unexpected in an FB model—of its point-like radio source [164], items to be discussed anon (tests 12, 13 and 16). Consistently, GW170817 had $E_{iso} = (5.4 \pm 1.3) \times 10^{46}$ erg, five orders of magnitude smaller than that of ordinary SHBs.

*9.3. The CB's Superluminal Velocity in SN-GRBs (Test 12)*

The first observation of an apparent superluminal velocity of a source in the plane of the sky was reported [165] in 1902, and since 1977 in many high resolution observations of highly relativistic jets launched by quasars, blazars, and micro-quasars. The interpretation of such observations within the framework of special relativity was provided in [166,167].

A source with a velocity $\beta c$ at redshift $z$, viewed from an angle $\theta$ relative to its direction of motion and timed by the local arrival times of its emitted photons has an apparent velocity in the plane of the sky:

$$V_{app} = \frac{\beta c \sin \theta}{(1+z)(1-\beta \cos \theta)} \approx \frac{\beta c \gamma \delta \sin \theta}{(1+z)}. \tag{22}$$

For $\gamma \gg 1$, $V_{app}$ has a maximum value $2 \gamma c / (1+z)$ at $\sin \theta = 1/\gamma$.

The predicted superluminal velocity of the jetted CBs cannot be verified during the prompt emission phase, because of its short duration and the large cosmological distances of GRBs. But the superluminal velocity of the jet in far off-axis, i.e. nearby low-luminosity GRBs, can be obtained from high resolution follow-up measurements of their AGs [168,169]. Below, two cases are treated in detail.

### 9.3.1. GRB980425

The radio and X-ray afterglow of GRB980425, the nearest observed SN-GRB with a known redshift, $z = 0.0085$, has so far offered the best opportunity to look for the superluminal signature of the highly relativistic jets which produce GRBs [168,169]. For some reason this has been totally overlooked in the late-time X-ray [132,133] and radio observations [170] of SN1998bw/GRB980425. But if the transient sources observed on days 1281 and 2049.19 from the direction of SN1998bw are the CB which produced GRB980425, this source moved by a viewing angle $\theta_s \simeq 2c/V_\perp \simeq 1/170$ rad with an average $V_{app} \simeq 340 c$. This interpretation implies that these sources are not present there anymore, and were not to be seen before SN1998bw/GRB980425 was observed.

Supportive CB-model evidence for the above values of $\theta_s$ and $V_{app}$ in GRB980425 is provided by other observations:

(i) The expected value of $E_p$, as given by Equation (2) with a typical $\epsilon_p = 1$ eV [12,53–57], is 58 keV. This is in good agreement with the observed [162]. $E_p = 55 \pm 21$ keV.

(ii) Recall that $R_g$ stands for the radius characterizing the extention of the glory surrounding an SN. A highly relativistic CB crosses it in a (local) time $R_g/c$. For an observer this corresponds to a GRB pulse of FWHM $\simeq 2(1+z)R_g/\gamma_0 \delta_0 c$. The averages of these widths and of their redshifts for ordinary GRBs are 0.89 s [170] and $\langle 1+z \rangle \simeq 3$. The ratio of these normalizing values to the FWHM $\simeq R_g \theta^2/c \simeq 2$ s duration of GRB980425 results in $\gamma \theta \simeq 9$. Upon substitution in Equation (20) one obtains $E_{iso}$(GRB980425) $\simeq 1.1 \times 10^{48}$ erg, in good agreement with its measured $(1.0 \pm 0.2) \times 10^{48}$ erg [162].

(iii) The 0.3–10 keV X-ray light-curve of the AG of GRB980425 [132,133,161] can be well fit by the CB model with $\gamma \theta \approx 8.7$, as shown in Figure 24.

9.3.2. GRB030329

As mentioned in Section 7, the CB model was used to fit the early AG of GRB030329 and to predict the discovery date of its associated SN, SN2003dh. This being a two-pulse GRB, the fits to its $\gamma$ rays and to its AG consequently involved two cannonballs. The prediction of the amount of their superluminal motion, based on the approximation of a constant ISM density, turned out to be wrong [171]. Subsequent observations of the AG showed a series of very clear re-brightenings, interpreted in the CB model as encounters of the CBs with ISM over-densities. Corrected by the consequent faster slow-down of the CBs' motion, the new CB-model results, were not a prediction, but were not wrong (see [99] and its Figure 2 for controversial details not mentioned here).

The relative proximity ($z = 0.1685$) of GRB030329 and its record-bright radio AG made possible its record long, high-resolution follow-up observations with the Very Long Baseline Array (VLBA) and Very Long Baseline Interferometry (VLBI), until 3018.2 days post GRB [172]. The earlier observers [173–176] adopted a circular Gaussian fit of the location of the radio source(s). They could not resolve separate images, except on day 51 after burst, when two sources $0.28 \pm 0.05$ mas ($\sim 0.80$ pc) away from each other were seen, with a statistical significance of $20\,\sigma$. By then the two sources (two CBs or a CB and the supernova?) had moved away from one another at an average relative apparent superluminal velocity $\langle V_{app} \rangle \approx 19\,c$. The "second component" was discarded:

> *"Since it is only seen at a single frequency, it is remotely possible that this image is an artifact of the calibration."* [171,173–176]

This second component, should it not be an artifact, would be, in the CB model, initially *hyperluminal*. An approximation to a detailed analysis would be the following. For an initial CB with $\gamma^2 \gg 1/\theta^2 \gg 1$ the apparent velocity of Equation (22) is $\approx 2\,c/\theta$. At later times, $t \approx t_b \approx 0.05$ days, when the CB has decelerated to the point that $[\gamma(t)\theta]^2 < 1$, $V_{app} \propto t^{-1/2}$ in a (rough!) constant-density approximation. The early *local* approximate apparent velocity is $V_{app}(t_b) \sim (1+z)\langle V_{app} \rangle (51\,\mathrm{d}/t_b)^{1/2} \approx 710\,c$.

A way to test whether a constant-density approximation for the bumpy ISM traversed by the CBs of GRB030329 is good—on average– is the following. At late times $\delta(t) = 2\,\gamma(t) \propto t^{-1/4}$. Consequently, $F_\nu \propto t^{-\alpha_\nu}\,\nu^{-\beta_\nu}$ with $\alpha_\nu = \beta_\nu + 1/2$, which are all well satisfied by the late-time X-ray afterglow [126] and radio observations of GRB030329 [173–176]. For example, the measured spectral index $\beta_X = 1.17 \pm 0.04$ in the 0.2–10 keV X-ray band [172], results in a late-time temporal decay index $\alpha_x = \beta_x + 1/2 = 1.67 \pm 0.04$, in good agreement with the observed $\alpha_x = 1.67$ [126] as shown in Figure 25.

The earlier VLBA measurements [171] could not resolve the separate images of SN2003dh and the CBs which produced GRB030329 and its AG. Assuming a disk shape, the radio image was fit with a circular Gaussian of diameter $2\,R_\perp(t)$. If this distance is a rough measure of the displacement of the second CB fired by SN2003dh, it corresponds to a mean superluminal velocity declining like $t^{-1/2}$, in agreement with the data, shown in Figure 26. The prediction is for $\gamma_0\,\theta = 1.76$, obtained for the observed [173–176] $E_{iso}(\mathrm{GRB030329}) = (1.86 \pm 0.08) \times 10^{52}$ erg and $\langle E_{iso}(\mathrm{O\ GRB}) \rangle \approx 7 \times 10^{52}$ erg for ordinary GRBs, assuming a standard candle approximation for GRBs and $\epsilon_p \simeq 1$ eV [12,53–57] in Equation (2). The observed late-time behavior, shown in Figure 26, suggests deceleration in a constant density ISM of a CB with $\gamma_0 \approx 400$.

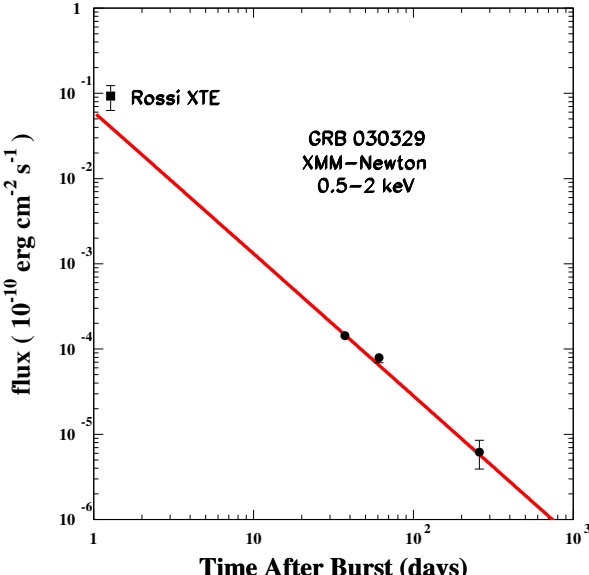

**Figure 25.** The late-time 0.5-2 keV X-ray AG of the joint source GRB030329/SN2003dh as measured by Rossi-XTE and XMM-Newton [177]. The line is the CB model expectation (for the predicted $\alpha_x = \beta_x + 1/2$) and, in this case, $\beta_x = 1.17 \pm 0.04$.

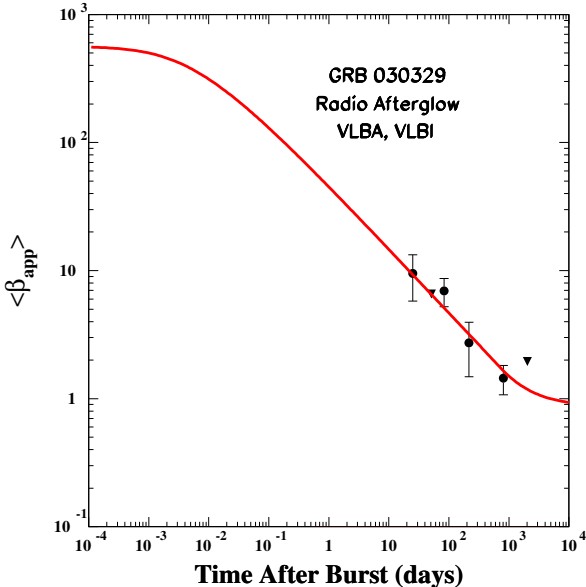

**Figure 26.** The time-averaged expansion rate of the radio image of GRB030329/SN2003dh [172]. The line is the predicted $\langle \beta_{app} \rangle$ of the CB, which produced GRB030329 and its AG, assuming that $2R_\perp$ is its distance from SN2003dh.

Initially, the radius of a CB in its rest frame increases at the speed of sound in a relativistic plasma, $c_s = c/\sqrt{3}$. At an observer's time $T$ the radius of a CB has increased to:

$$R(T, \theta) \approx \frac{c_s}{(1+z)} \int_0^T \delta(t, \theta) \, dt, \tag{23}$$

where use has been made of the relation between $T$ and the time in the CB's rest system. When the first radio observations took place, $T = 2.7$ days after burst, the result for this GRB—for the parameters of the CB-model description of its AG and the deceleration law of Equation (9)—is $R(T) \approx 5.7 \times 10^{17}$ cm. This is more than an order of magnitude larger than the largest source size that could have resulted in diffractive scintillations.

The FB model has been used to provide successive a posteriori interpretations of. the observed superluminal expansion [172] of the image size of the source of the radio AG of the GRB030329-SN2003dh pair, as the observations progressed. All of them were parametrizations with many adjustable parameters. A re-brightening of the source was expected in the FB model as the counter jet became non-relativistic [178,179]. But no re-brightening was detected up to 10 years after the burst [177]. As stated by the discoverers of the "second source" [171]: *This component requires a high average velocity of 19c and cannot be readily explained by any of the standard models.*

## 10. Fast Extragalactic X-ray Transients, Tests 14 & 15

In the CB model GRBs produce two main types of extragalactic X-ray transients: X-ray flashes , which are narrowly beamed LGRBs viewed far off axis [180], and fast X-ray transients (XRTs) which are emissions from a wind nebula powered by a newly-born millisecond pulsar in a binary neutron-star merger and in SN-less LGRBs [180].

As stated earlier, GRB pulses produced by highly relativistic CBs and observed from a far off-axis angle $\theta \gg 1/\gamma$, appear as XRFs [180] with $L_p$, $E_{iso}$ and $E_p$ reduced by a factor $\approx [1/\gamma^2(1-\cos\theta)]^{4,3,1}$, respectively, and a duration T(FWHM) larger by a factor $\approx \gamma^2(1-\cos\theta)$, all of it in comparison with ordinary GRBs viewed from angles $\theta \sim 1/\gamma$. The strong reduction of the peak luminosity $L_p$ of far off-axis GRBs makes them visible only from relatively small $z$. The orphan isotropic X-ray AGs of far off-axis (unobserved SGRBs [180]) are much longer and are visible up to very large cosmological distances.

The X-ray transients XRT 000519 [77] and XRT 110103 [181] could have been classified as XRFs. The correlation of Equation (5) for XRFs (far off axis GRBs [180]), shown in Figure 5, is well satisfied by $E_p \approx 1.5 \pm 0.5$ keV and $E_{iso} \approx (4 \pm 2) \times 10^{44}$ erg, obtained [77] for XRT 000519, as shown in Figure 27. This is quite impressive, since the the $E_p$ of XRT 000519 is more than three orders of magnitude lower than that of GRB 980425, the next low-$E_p$ record holder in the figure.

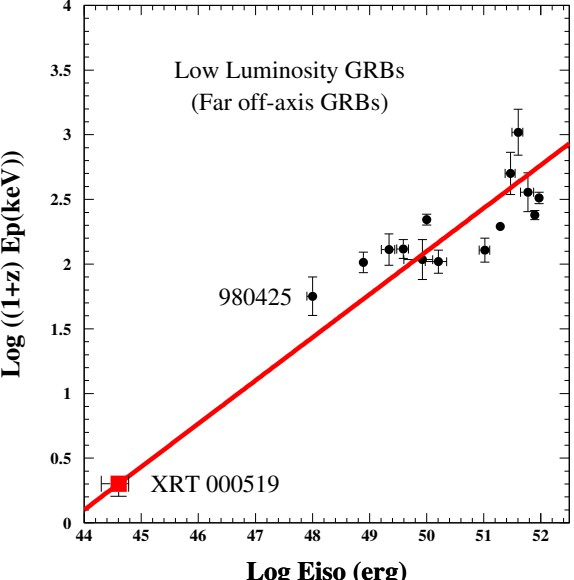

**Figure 27.** The $[E_p, E_{iso}]$ correlation in LGRBs viewed far off axis (which include low-luminosity LGRBs and XRFs), with the addition of XRT 000519 [77] (full red square). The line is the CB model predicted correlation, Equation (5).

To test further the contention that XRT 000519 [77] and XRT 110103 [181] can be interpreted as far off-axis LGRBs, one may fit their pulse shapes (count rate as function of time) by the approximate CB-model shape of far off-axis GRBs following from Equation (6)

$$\frac{dN_\gamma(E > E_m)}{dt} \propto \frac{t^2}{(t^2 + \Delta^2)^2} \, e^{-E_m/E_p(0)} \left[1 - \frac{t}{\sqrt{t^2 + \tau^2}}\right], \tag{24}$$

where $E_m$ is a minimum energy of detection, $E_m = 0.3$ keV for the two pulses of XRT000519 shown and fit in Figures 28 and 29. The single pulse of XRT110103 is fit with Equation (24) in Figure 30.

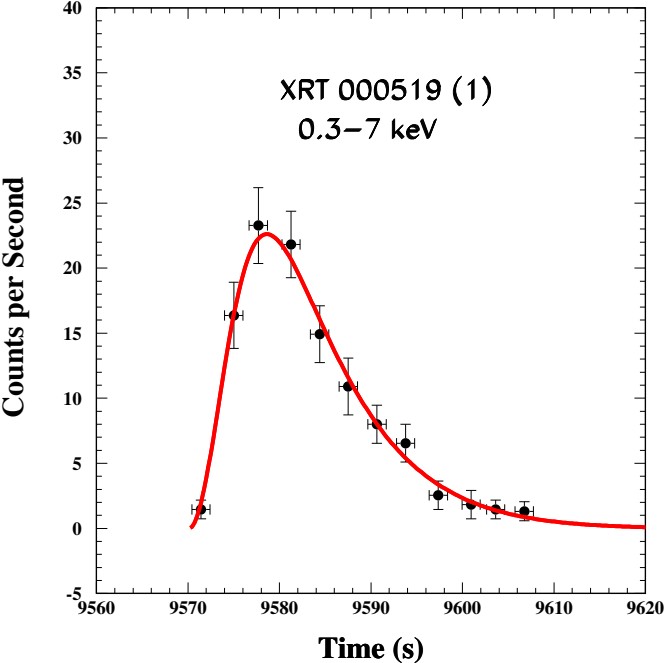

**Figure 28.** The best fit pulse shape given by Equation (24) for the first pulse of XRT 000519, reported in [77]. The fit parameters are $\Delta = 9.65$ s, $\tau = 17.50$ s, $E_p(0) = 13.70$ keV (for time $t$ since 9570.10 s); $\chi^2/\mathrm{dof} = 0.39$.

In the CB model the early time X-ray AGs of both near or far off-axis SGRBs are hypothesized to be powered by a newly born millisecond pulsar with a braking index $n \approx 3$. Their energy flux satisfies Equation (19). In Figure 31 the X-ray light curve of the fast extragalactic transient CDF-S XT2 [182] is shown and compared to the early-time observations of the X-ray AGs of all SGRBs well sampled at early times and reported in the Swift-XRT Light Curves Repository [95,96].

*XRFs in the FB Model*

In fireball models XRFs are assumed to belong to a different class of GRBs. But their consequently estimated yearly rate over the whole sky is negligible compared to the rate of XRFs extracted from observations: $\sim 1.4 \times 10^5$ per year over the whole sky [181].

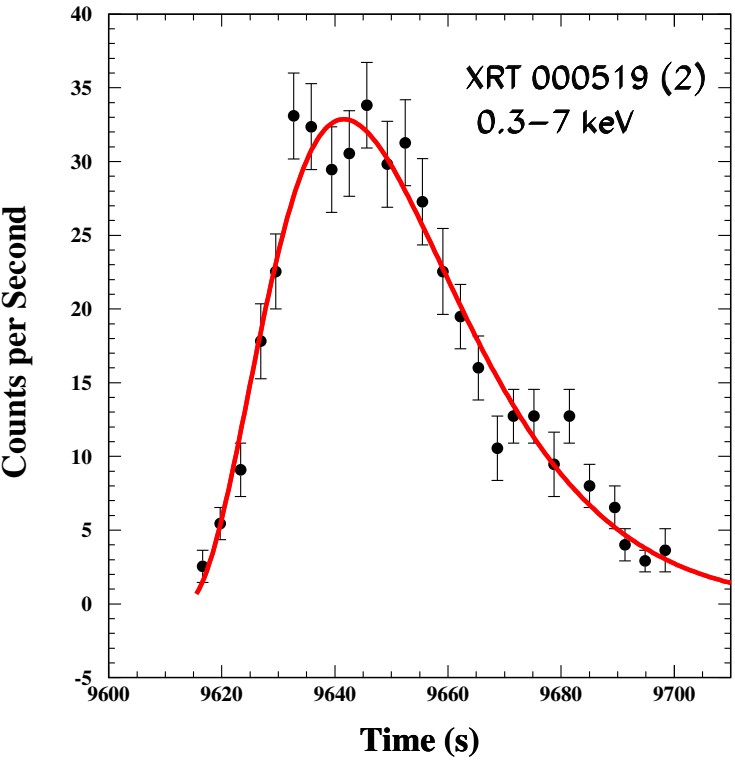

**Figure 29.** The light curve of the second pulse of XRT 000519 [77] and the CB-model's predicted shape of Equation (24) with the best-fit parameters $\Delta = 35.54$ s, $\tau = 28.58$ s, $E_p(0) = 27.11$ keV (for time $t$ since 9613.28 s); $\chi^2/\text{dof} = 1.28$.

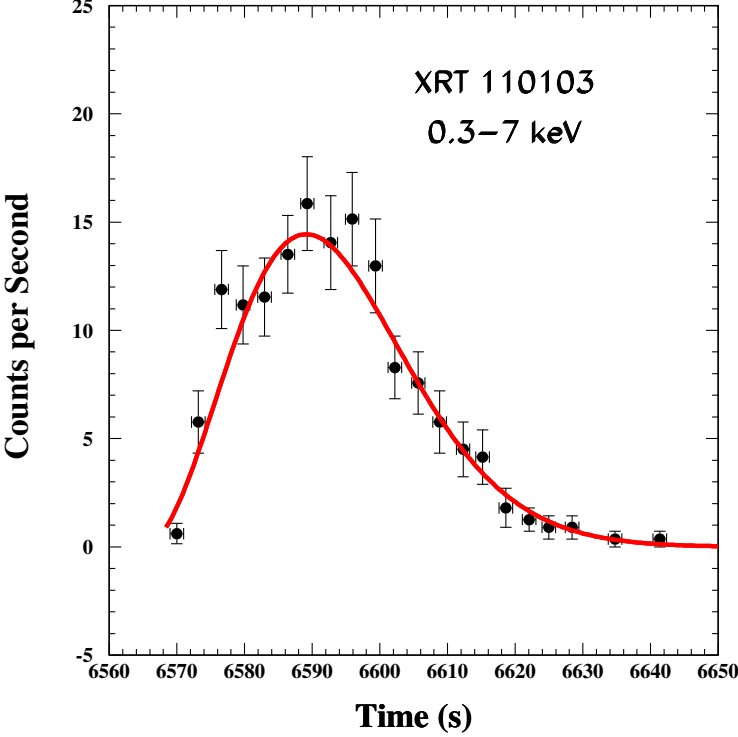

**Figure 30.** The pulse shape of XRT 110103 [181] and the CB-model shape of Equation (24) with the best fit parameters, $\Delta = 45.03$ s, $\tau = 23.16$ s, $E_p(0) = 12.18$ keV (for time $t$ since 6564.90 s); $\chi^2/\text{dof} = 1.20$.

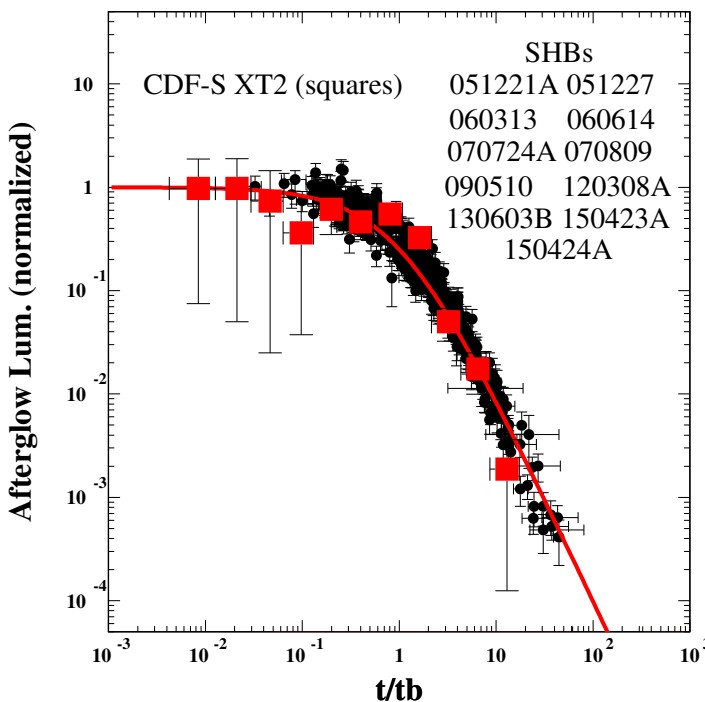

**Figure 31.** Comparison between the scaled 0.3-10 keV light curves of the well sampled X-ray AGs of SGRBs during the first couple of days after burst measured with the Swift XRT and the 0.3 to 10 keV light curve of CDF-S XT2 [182]. The line is the universal behavior of Equation (19) for a PWN AG powered by a newly-born millisecond pulsar with a braking index $n = 3$.

## 11. GRB Theories Confront SHB170817A (Test 16)

This event is an optimal case to close the discussion of the comparisons between different models of GRBs.

GW170817 was the first binary neutron-star merger detected with Ligo–Virgo [30–33] in gravitational waves (GWs). It was followed by SHB170817A, $1.74 \pm 0.05$ s after the end of the GW's detection, with an afterglow across the electromagnetic spectrum, which was used to localize it [183] to the galaxy NGC 4993. The GW170817/SHB170817A association was the first indisputable confirmation that pairs of neutron stars merging due to GW emission produce GRBs. Earlier, other mechanisms of GRB emission in such mergers had been suggested: in 1984 the explosion of the lighter NS after tidal mass loss [6], later called a "Macronova" event [7] and, in 1987, a neutrino annihilation fireball [8] around the remnant neutron star. In 1994, the fireball mechanism was replaced [12] by ICS of external light by a highly relativistic jet of ordinary plasma launched by fall back ejecta on the remnant star after the merger, which became the basis of the CB model of SGRBs and their afterglows.

The relative proximity of SHB170817A provided many critical tests of SGRB theories. As mentioned in Section 8, two days before the GW170817/SHB170817A event, the CB model of GRBs was used to predict [134] that most of the SGRBs associated with Ligo–Virgo detections of NS/NS mergers would be beamed far off-axis. Consequently, only a small fraction of them would be visible as low-luminosity far off-axis SGRBs [134]. Additionally [134], the early afterglow, powered by the spin down of the newly born millisecond pulsar, would generate a characteristic isotropic universal light-curve [34], visible from all SHBs associated with gravitational wave detections of neutron star mergers.

Shortly after the detection of SHB170817A, several authors claimed [27] that it was an ordinary near-axis SGRB, even though its $E_{iso}$ was four orders of magnitude smaller than that of typical SGRBs. Others, who claimed in the past that ordinary and low luminosity GRBs belong to two different classes, suggested that SHB170817, was ordinary, but viewed far off-axis. Moreover, other changes—labeled "structured jets"—were introduced. Despite the availability of many adjustable parameters, all such models failed to predict correctly

the future evolution of the AG light curves of SHB170817A, even further readjusting multiple parameters to best fitting the entire earlier data, as shown later.

### 11.1. The Properties of the SHB170817A Ejecta

#### 11.1.1. A Superluminally Moving Source

The VLBI/VLBA observations of the radio AG [163] of SHB170817A provided images of an AG source escaping from the GRB location with superluminal celerity. Such a behavior in GRBs was predicted by the CB model [12,53–57] two decades ago [163]. Figure 32, borrowed from [163], shows a displacement with time of a point-like radio source—as seen before in micro-quasars and blazars [12,53–57]—, rather than an unresolved image of a GRB and its AG expanding with a superluminal speed, as claimed before in the case of GRB030329 [173–176]. The Figure displays the angular locations of the radio AG of SHB170817A moving away in the plane of the sky from the SHB location by $2.68 \pm 0.3$ mas [163] between day 75 and day 230. In [163] this image is called "a jet".

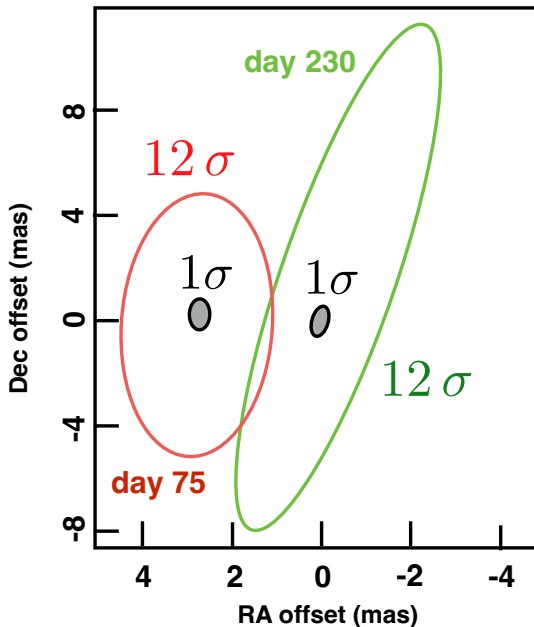

**Figure 32.** Proper motion of the radio counterpart of GW170817 [163] The centroid offset positions (shown by $1\sigma$ error bars) and $12\sigma$ contours of the radio source detected 75 d (red) and 230 d (green) post-merger with VLBI at 4.5 GHz. The radio source is consistent with being unresolved at both epochs. The proper motion vector of the radio source has a magnitude of $2.7 \pm 0.3$ mas. The $1\sigma$ domains have been grey-colored not to deemphasize the effectively point-like nature of the source (a CB).

#### 11.1.2. Superluminal Motion

Consider a highly relativistic ($\beta \approx 1$) CB. Its apparent velocity, Equation (22), can also be expressed in terms of observables:

$$V_{app} \approx \frac{c \, \sin\theta}{(1+z)\,(1-\cos\theta)} \approx \frac{D_A \, \Delta\theta_s}{(1+z)\Delta t}, \tag{25}$$

where $\Delta\theta_s$ is the angle by which the source is seen to have moved in a time $\Delta t$. The angular distance to SHB170817A to its host galaxy NGC 4993, at $z = 0.009783$ [183], is $D_A = 39.6$ Mpc, for the local value $H_0 = 73.4 \pm 1.62$ km/s Mpc obtained from Type Ia SNe [184,185]. The location of the VLBI-observed source—which moved $\Delta\theta_s = 2.7 \pm 0.3$ mas in a time $\Delta t = 155$ d (between days 75 and 230)—implies $V_{app} \approx (4.0 \pm 0.4)\, c$, which, solving for the viewing angle $\theta$ in Equation (25), results in $\theta \approx 27.8 \pm 2.9$ deg. This value agrees

with $\theta = 25 \pm 8$ deg, obtained [186,187] from the gravitational wave observations [30–33] for the same $H_0$ [184,185] and the hypothesis that the CB was ejected along the rotation axis of the binary system.

### 11.1.3. Initial Lorentz Factor

In the CB model SGRBs, much as LGRBs, can be treated as approximately standard candles viewed from different angles. They satisfy similar correlations. In particular low luminosity (LL) SHBs such as SHB170817A are ordinary (O) SHBs viewed far off-axis. Consequently, their $E_{iso}$ and $E_p$ are expected to obey the relations

$$E_{iso}(\text{LL SHB}) \approx \langle E_{iso}(\text{O SHB}) \rangle / [\gamma^2 (1 - \cos\theta)]^3 , \tag{26}$$

$$(1 + z) E_p(\text{LL SHB}) \approx \langle (1 + z) E_p(\text{O SHB}) \rangle / [\gamma^2 (1 - \cos\theta)]. \tag{27}$$

Given the measured value $E_{iso} \approx 5.4 \times 10^{46}$ erg of SHB170817A [170], the mean value $\langle E_{iso} \rangle \approx 1 \times 10^{51}$ erg of ordinary SGRBs, and the viewing angle $\theta \approx 28$ deg obtained from the observed superluminal velocity of the source of its radio afterglow [163], Equation (27) yields $\gamma_0 \approx 14.7$ and $\gamma_0 \theta \approx 7.2$.

### 11.1.4. Prompt Emission Observables

The CB model correlations of Equations (3) and (4) are well satisfied by each of the three major types of GRBs: SN-LGRBs, SN-less LGRBs and SN-less SGRBs, as was demonstrated in Figures 3–5. Equations (26) and (27), for the viewing angle of the CB obtained from its apparent superluminal motion [163], result in the following additional tests of CB model predictions:

**Peak energy.** Assuming that SHBs have the same redshift distribution as GRBs (with a mean value $\langle z \rangle \approx 2$), and given the observed $\langle E_p \rangle = 650$ keV of SHBs [27], one obtains $\langle (1+z) E_p \rangle \approx 1950$ keV. Consequently, Equation (27) with $\gamma_0 \theta \approx 7.2$ and $z \approx 1$ yields $E_p \approx 75$ keV for SHB170817A. This is to be compared with $E_p = 82 \pm 23$ keV ($T_{90}$) reported in [25], $E_p = 185 \pm 65$ keV estimated in [27], and $E_p \approx 65 + 35(-14)$ keV estimated in [28], from the same data, with a mean value $E_p = 86 \pm 19$ keV, agreeing with the expectation.

**Peak time.** In the CB model the peak time $\Delta t$ after the beginning of a GRB or SHB pulse is roughly equal to half of its FWHM. Assuming again that SHBs are roughly standard candles, the dependence of their $\Delta t$ values on the viewing angle $\theta$ is

$$\Delta t(\text{LL SHB}) \approx \gamma_0^2 (1 - \cos\theta) \langle \Delta t(\text{O SHB}) \rangle, \tag{28}$$

For $\theta \approx 28$ deg, obtained from the superluminal motion of the source of the radio AG of SHB170817A, $\Delta t \approx 0.58$ s obtained from the prompt emission pulse of SHB170817A (see Figure 9), and $\langle \text{FWHM(SHB)} \rangle = 55$ ms, Equation (25) yields $\gamma_0 \approx 14.7$. Using Equations (26) and (27), and $\gamma_0 \theta \simeq 7.2$ one checks that this value of $\gamma_0$ is consistent with $E_{iso} = 5.4 \times 10^{46}$ estimated in [27], and $\langle E_p \rangle = 86 \pm 19$ keV the mean of the estimates in [34].

Moreover, in the CB model the shape of resolved SHB and GRB pulses satisfies $2\,\Delta t \approx \text{FWHM} \propto 1/E_p$. Using the observed $\langle \text{FWHM(SHB)} \rangle \approx 55$ ms, $\gamma_0 \sim 14.7$, and $\theta \approx 28$ deg, Equation (28) for SHB170817A results in $\Delta t \approx 0.63$ s, in good agreement with its observed value, $0.58 \pm 0.06$ s[2].

### 11.2. The Single Pulse's Correlation between Energy and Time

In the CB model the peak-time of a pulse, $T_p \propto (1+z)/\gamma_0 \delta_0$, and its peak energy $E_p$ of Equation (2) are properties of resolved or isolated pulses, which occur in a large fraction of SHBs, not GRBs. SHB170817A, being a one-peak event, is a good case to study this variable. One of the simplest CB-model predictions is the $[E_p, T_p]$ correlation

$$E_p \propto 1/T_p . \tag{29}$$

In Figure 33 this correlation is compared with the values of $E_p$ and $T_p$ in the GCN circulars for resolved SGRB pulses measured by the Konus-Wind and by the Fermi-GBM collaborations. The figure shows that the prediction is satisfied by most of the measurements, in particular the ones with small error bars.

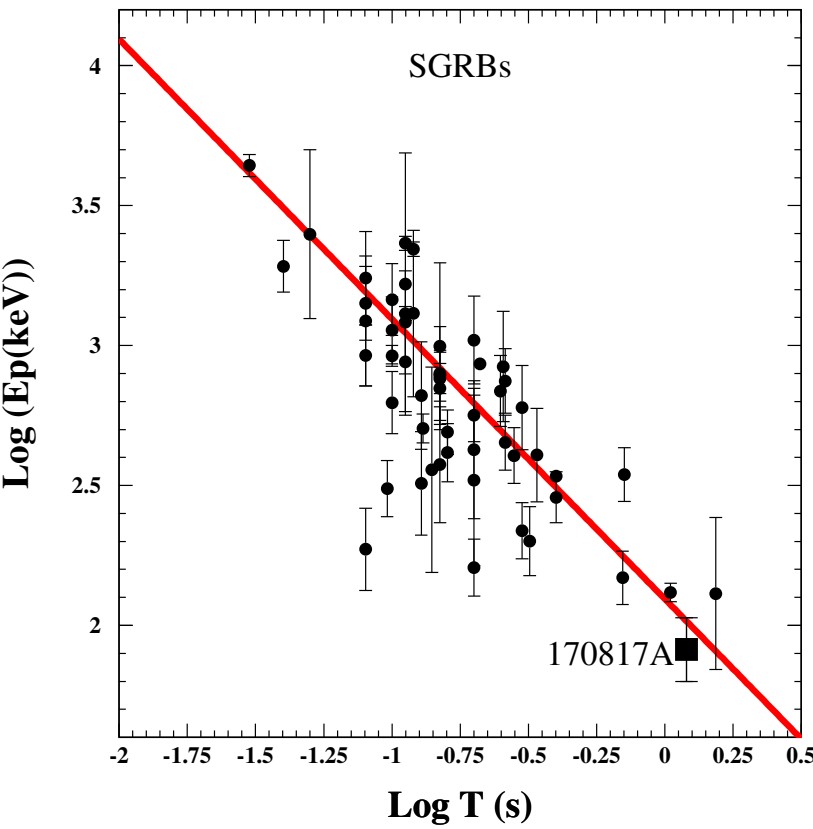

**Figure 33.** Comparison between the predicted $[E_p, T_p]$ correlation of Equation (29) and the corresponding data in GCN circulars for 54 resolved pulses of SGRBs, obtained by the Konus-Wind and Fermi-GBM collaborations.

11.2.1. The Early Time Afterglow

The bolometric AG of SHB170817A, during the first few days after burst, has the universal shape of the early-time X-ray AGs of all SGRBs and SN-less LGRBs well sampled during the first few days after the prompt emission. This universal shape is the one already shown for SGRBs in Figure 21. In Figure 34, it is shown for the bolometric light curve [118] of SHB170817A during the first two weeks after burst. In the absence of reliable observational information on the environments of compact NS/NS and NS/BH binaries shortly before they merge, and glory of optical-energy photons, similar to that adopted in the CB model of LGRBs, was erroneously hypothesized for SHBs as well. It has been replaced by a soft X-ray glory once the superluminal motion of the ejected CB in SHB170817 was measured [163].

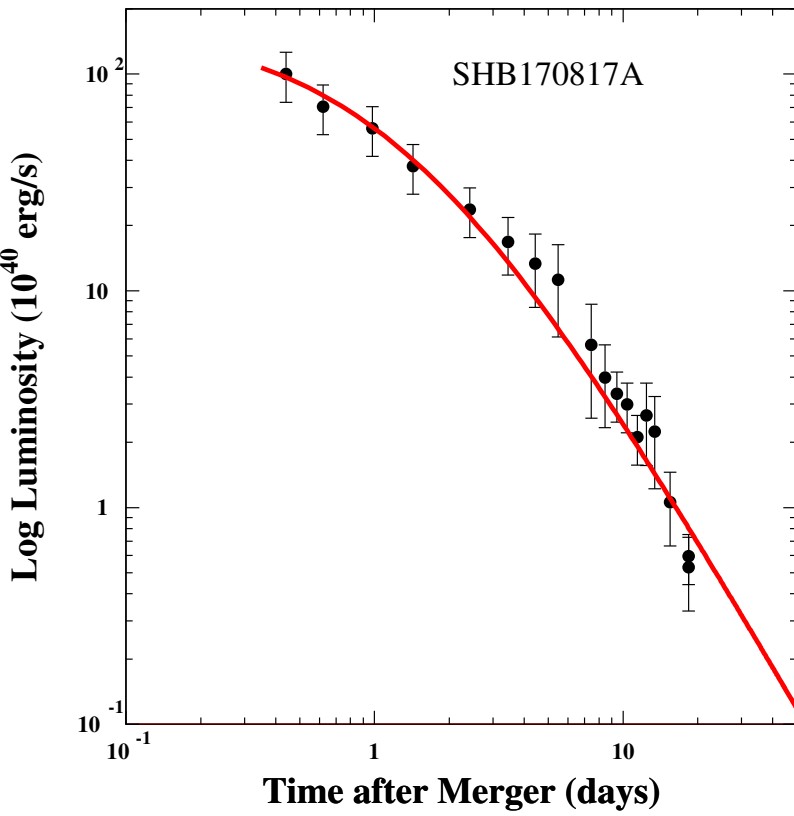

**Figure 34.** Comparison between the observed [118] bolometric light curve of SHB170817A and the universal light curve of Equation (19), assuming the presence of a milli-second pulsar with $L(0) = 2.27 \times 10^{42}$ erg/s and $t_b = 1.15$ d. The fit has $\chi^2/\text{dof} = 1.04$.

### 11.2.2. The Late-Time Afterglow

As long as the Lorentz factor of a decelerating CB is such that $\gamma^2 \gg 1$, $\gamma\,\delta \approx 1/(1-\cos\theta)$ and the spectral energy density of its unabsorbed synchrotron AG—Equation (8)—can be rewritten as

$$F_\nu(t, \nu) \propto n(t)^{\beta_\nu + 1/2} \left[\gamma(t)\right]^{2\beta_\nu - 4} \nu^{-\beta_\nu} \tag{30}$$

with $n(t)$ the baryon density of the medium encountered by the CB and $\beta_\nu$ the spectral index of the emitted synchrotron radiation.

For a constant density, the deceleration of the CB results in a late-time $\gamma(t) \propto t^{-1/4}$ [94], and as long as $\gamma^2 \gg 1$,

$$F_\nu(t, \nu) \propto t^{0.72 \pm 0.03} \nu^{-0.56 \pm 0.06}, \tag{31}$$

with use of the observed [164] $\beta_\nu = 0.56 \pm 0.06$, which extends from the radio (R) band, through the optical (O) band, to the X-ray band.

If the CB moved out from within a cloud of constant internal density into a wind-like density distribution (proportional to $r^{-2}$) its deceleration rate diminished and $\gamma(t)$ became practically constant. Consequently, the time dependence of $F_\nu$ in Equation (30) becomes a fast decline described by

$$F_\nu(t, \nu) \propto t^{-2.12 \pm 0.06} \nu^{-0.56 \pm 0.06}. \tag{32}$$

These CB-model approximate rise and fall power-law time dependences of the light curves of the ROX afterglow of SHB170817, with temporal indices $0.72 \pm 0.03$ and $-2.12 \pm 0.06$, respectively, are in good agreement with the power-law indices extracted in [164], $0.78 \pm 0.05$ and $-2.41 + 0.26/-0.42$, respectively, from a phenomenological parametrization of the measured radio light curves of SHB170817 [164] during the first year

after burst. They also agree with the indices subsequently extracted in [188], $0.86 \pm 0.04$ and $-1.92 \pm 0.12$, as shown in Figure 35.

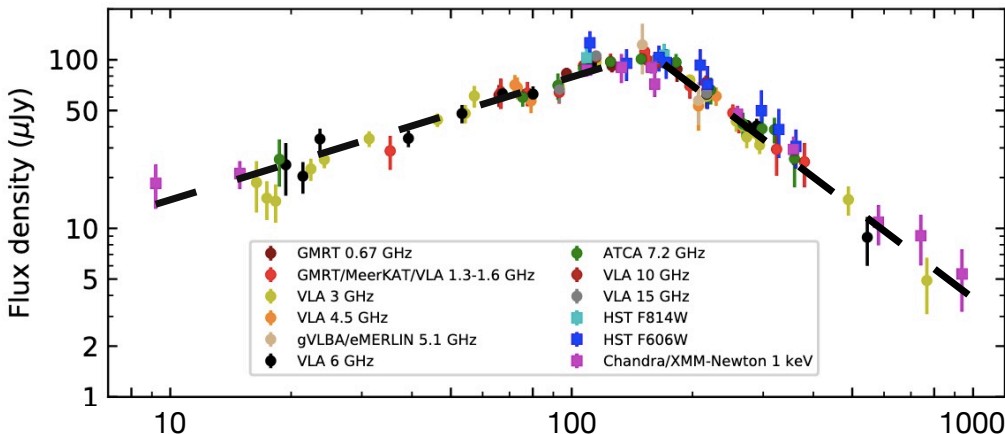

**Figure 35.** Adapted from Figure 2 of [188]. The radio light curve of SHB170817A, measured until 940 days post-merger, spanning multiple frequencies, and scaled to 3 GHz using the spectral index $-0.584$. The early-time trend expected in the CB model is the rising black-dashed line. The late-time trend, also black-dashed, is for an assumed $1/r^2$ ISM density decline.

## 12. FB Model Interpretations of SHB170817A

Soon after the discovery of the late-time radio, optical and X-ray afterglows of SHB170817A, many FB model best fits to these initially rising light curves were published. They involved many choices and best-fit parameters (e.g., [189] and references therein). When new observations were obtained, they did not follow the FB models' predictions. They were replaced with "postdictions" and a readjusting of assumptions and parameters. These repeated failures illustrate the flexibility of the FB models, rather than their validity. Some examples are summarized below.

In November 2017 the numerous authors of [189] concluded that *The off axis jet scenario as a viable explanation of the radio afterglow of SHB170817A is ruled out* and that a choked jet cocoon is most likely the origin of the gamma rays and rising AG of SHB170817A. Their claim was based on a best fit (Figure 2 in [189]) to the 3 GHz radio observations obtained with ATCA and VLA before November 2017.

In April 2018, the observers among the authors of [190] reported that their 2–9 GHz radio observations of GW170817 covering the period 125–200 days post-merger, taken with the Australia Telescope Compact Array and the Karl G. Jansky Very Large Array, unexpectedly peaked at day $149 \pm 2$ post merger and declined thereafter. RXO observations of the AG were continued until two years after burst. They are shown in Figure 35 (Figure 2 of [164]). The parametrization is a smoothly broken power-law [116] with a temporal index $\alpha = 0.84 \pm 0.05$ on the rise, peak time $149 \pm 2$ day, and a temporal index $1.6 \pm 0.2$ on the decay. The authors of [164] reached conclusions opposite to their earlier ones [189,190] and to their previous arXiv versions. To wit, in [164] they have reported a *strong jet signature in the late-time light-curve of GW170817.* They justified their new conclusion by the fact that the post break flux density parametrized as $F_\nu(t) \propto t^\alpha \nu^\beta$ yields $\beta = -0.54 \pm 0.06$ and $\alpha = -2.17$, consistent with the FB model prediction $\alpha = -p$ post break with $p$ the power-law index of the energy distribution of the radiating electrons [191].

The cited FB model interpretations are not self-consistent for various reasons:

(a) The relation $\alpha = -p$ is only valid for a conical jet with a fast lateral expansion ($V_\perp \approx c$) that has stopped propagating after the jet break time [191]. The fast spreading and the stopped propagation of the jet are not supported by the VLBI observations of the radio AG of SHB170817A [163], which show a superluminal compact source (a CB), rather than the cited features.

(b) The relation $\alpha = -p$ post break is seldom satisfied by GRB AGs and often yields $p < 2$.

(c) Due to rather large measurement errors, it is not yet clear that the temporal behavior of the AG of SHB170817A after break can be well parametrized by a broken power-law satisfying $\alpha = -p$.

(d) All types of FB models—with conical or structured jets—used to fit the multi-band afterglow of SHB170817A failed to correctly predict the subsequent data. This is demonstrated, for example, by the arXiv versions 1–4 of [192] where the evolution of the AG was first incorrectly predicted by a structured jet with a relativistic, energetic core surrounded by slower and less energetic wings, propagating in a low density ISM, as shown in Figure 36. As soon as the AG break around day 150 and its subsequent fast decline were observed, the structured jet model with its dozen or so adjustable parameters had no problem to accommodate this behavior, see Figure 37.

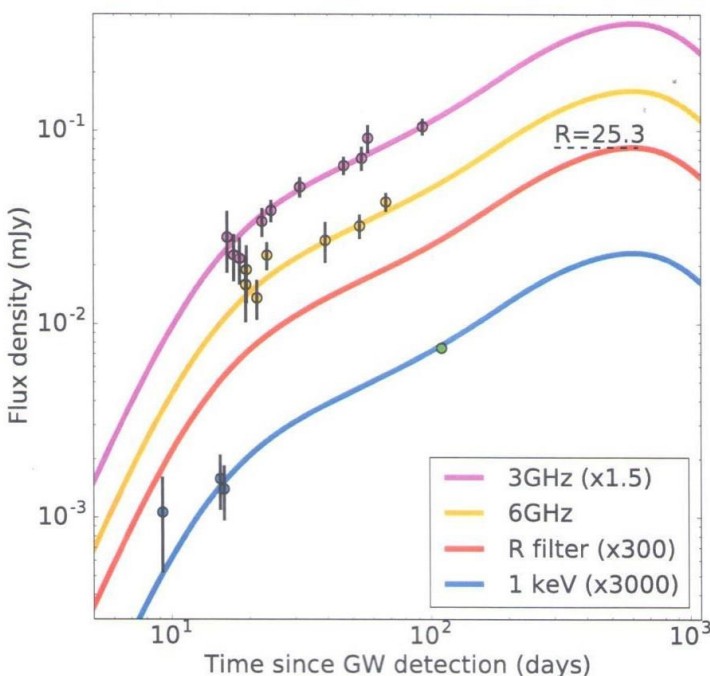

**Figure 36.** Best fit light curves of an off-axis structured jet model [192] to the ROX AGs of SHB170817A measured before December 2017 (Figure from the first version of [192] posted in the arXiv on 8 December 2017).

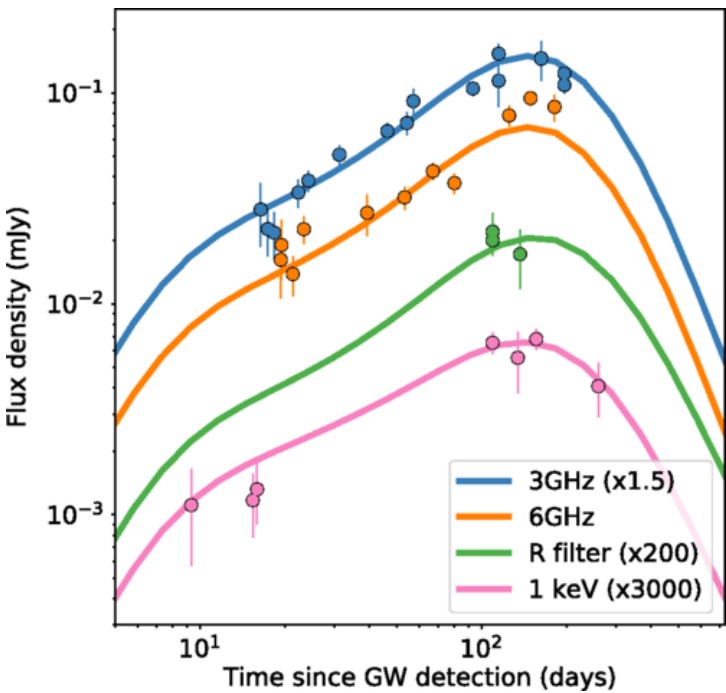

**Figure 37.** Best fit light curves to the ROX AG of SHB170817A up to April 2018, obtained from a structured jet model [192]. Reported in version 4 of [192] posted in the arXiv on 11 May 2018.

## 13. Conclusions

Table 3 contains a summary of many observational tests of the CB and FB models. Even without insisting on the cases for which the CB model provided successful *predictions*, and without reference to Table 4 on the evolution of the FB model(s), the conclusions are not new, but they should be overwhelmingly clear.

**Table 3.** Critical Tests of The Cannonball and Fireball models of GRBs and SHBs.

| Test | Cannonball Model | | Fireball Model | |
|------|------------------|---|----------------|---|
| Test 1 | Large GRB linear polarization | √ | Small GRB polarization | X |
| Test 2 | Prompt emission correlations | √ | Frail relation | X |
| Test 3 | Inverse Compton GRB pulses | √ | Curvature-shaped pulses | X |
| Test 4 | SN-GRBs: Canonical afterglow | √ | Canonical AG not expected | X |
| Test 5 | AG's break correlations | √ | AG's Break correlations | X |
| Test 6 | Post-break closure relation | √ | Post-break closure relation | X |
| Test 7 | Missing breaks (too early) | √ | Missing breaks (too late) | X |
| Test 8 | Chromatic afterglow | √ | Achromatic afterglow | X |
| Test 9 | MSP-powered AG of SN-less GRB | √ | Magnetar jet re-energization | X |
| Test 10 | GRB rate ∝ SFR | √ | GRB rate not ∝ SFR | X |
| Test 11 | LL GRBs = far off-axis GRBs | √ | LL GRBs = Different GRB class | X |
| Test 12 | Super-luminal CBs | √ | Superluminal fireball | X |
| Test 13 | SHBs optical AG powered by NS | ? | SHBs + macronova | ? |
| Test 14 | XRFs = Far off-axis LGRBs | √ | Different class of LGRBs | X |
| Test 15 | XRTs = NS-powered AGs | √ | AGs of Far-off-axis GRBs | X |
| Test 16 | Radio image of SHB170817A: a CB | √ | A complex structured jet | X |

**Table 4.** Majority and minority views on GRBs preceding decisive observations.

| Key Property | Majority View | | Minority View | |
|---|---|---|---|---|
| Location: | Galactic | X | Extragalactic | √ |
| Produced by | Relativistic $e^+e^-\gamma$ fireball | X | Highly relativistic plasmoids | √ |
| Production mechanism | Collisions of $e^+e^-$ shells | X | ICS of light by plasmoids (CBs) | √ |
| Prompt Emission | Synchrotron radiation (SR) | X | Inverse Compton scattering | √ |
| GRB geometry | Isotropic | X | Very narrowly beamed | √ |
| LGRBs origin | Stellar collapse to BH | X | Stripped-envelope SN | √ |
| Afterglows' origin | SR from shocked ISM | X | Synchrotron from CBs | √ |
| Afterglows' geometry | Isotropic | X | Narrowly beamed | √ |
| SN1998bw/GRB980425 | Rare SN/Rare GRB | X | SNIc-GRB viewed far off-axis | √ |
| LL GRBs | Different class of GRBs | X | Normal GRBs seen far off-axis | √ |
| SN-Less LGRBs | Stellar Collapse to BH | ? | Phase Transition in HMXRBs | ? |
| AG plateau origin | Jet re-energization | X | Early time jet deceleration | √ |
| AG break origin | Deceleration of conical jet | X | Deceleration of CBs | √ |
| Missing jet breaks | Too late to be seen | X | Too early to be seen | √ |
| Observed rate of GRBs | $\propto$ SFR + evolution | X | $\propto$ SFR, modified by beaming | √ |
| Geometry | Spherical $\rightarrow$ Conical shells | X | Succession of cannonballs | √ |

**Author Contributions:** All authors contributed equally. All authors have read and agreed to the published version of the manuscript.

**Funding:** S. Dado and A. Dar had no funding support. A. De Rujula has received funding/support from the 759 European Unions Horizon 2020 research and innovation programme under the Marie Sklodowska- 760 Curie grant agreement No 860881-HIDDeN.

**Data Availability Statement:** Data presented in this study whose source is not explicitly reported are available on request from the authors.

**Acknowledgments:** ADR acknowledges that this project has received funding/support from the European Union's Horizon 2020 research and innovation programme under the Marie Sklodowska-Curie grant agreement No 860881-HIDDeN.

**Conflicts of Interest:** The funders had no role in the interpretation of data nor in the decision to publish the results.

## Notes

[1] The original assumption in the CB model was that the interactions between a CB and the ISM were elastic. It was later realized, in view of the shape of AGs at late times, that a plastic collision—wherein most of the intercepted ISM is engulfed by the CB—was a better approximation in the AG phase.

[2] Quite obviously, the replacements of physical parameters by their means may not be completely reliable, not only because of the spread in their values, but also because of detection thresholds and selection effects.

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
