# Peer review of "Critical Tests of Leading Gamma Ray Burst Theories"

_universe, doi:10.3390/universe8070350_

Round 1

Reviewer 1 Report

The paper is well written and provides a comprehensive evaluation of the successes of the CB model compared to the FB model. I only found two minor errors which should be corrected.

1) Figure 16 has several yellow stars which are not identified in the caption.  These should be identified. I also suggest not using yellow because it is difficult to see.Pick a different color (green?).

2) Page 43 line 716 chocked -> choked

Author Response

We thank the referee for his report and comments.

  1. We have replaced Figure 16 with a new Figure 16.
  2. Page 43 line 716 chocked has been corrected to choked

Reviewer 2 Report

In their manuscript "Critical Tests of Leading Gamma Ray Burst Theories" the authors
review predictions of two competing models of of the still most enigmatic and violent
phenomena of astroparticle physics, the gamma ray burs. In particular, the authors
promote, proposed by them earlier, the theory of cannon ball origin of the prompt and afterglow emission
of gamma ray bursts. The cannon ball model looks quite natural,
once collapsar ejects plasmoids, probably akin a cumulative effect of ejection of blobs
of matter at extremely  violent event of a SN explosion, in case of long GRBs or
merging of neutron stars in case of short gamma ray burst, the prompt GRB emission as well as afterglow emission are generated along with propagation of the cannon balls trough the ambient light and interstalar medium. Thus the mechanism looks quite universal requiring not too many parameters. The tests of the mechanism are
discussed in great details in the manuscript, all ideas discussed in the manuscript are very well crystallized. With no doubts, I recommend the review for publication in the MDPI Universe.

Some remarks for a revision are presented bellow.

Line 200 -  the definition "∆ is approximately the peak time of the pulse in the observer’s frame –originating at the time when the CB becomes transparent to its internal radiation" is not
quite clear for a reader, in particular the wording  "its internal radiation"

Line 211 - to be edited as "full width at half maximum (FWHD)"

Line 241 - (AG) has been defined earlier, should be removed

Line 258 - The authors apply therm "plastic" collision, is it the same as inelastic?
            Also, are there microphysical (firs principal)
            arguments in favor of inelastic collision inside the cannon ball.

Fig.16 - The figure would me more convincing if it exhibited also the uncertainties
          of predictions of the post-break index by CB and FB models, since
          it is said that the statistical qualities of both fits are similar.

Author Response

We thank the referee for his kind review and useful remarks:

  1. Line 200 {currently 2003} - "becomes transparent to its internal radiation"  was replaced with "becomes transparent to inversee Compton scattering"
  2. Line 241 {currently 245} - the repeated definition of AG has been removed.
  3. Line 258  Concerning plastic collision: Footnote 1, where "plastic collision "  first appears, it has been modified to read  "The original assumption in the CB model was that the interaction between the CB and the ISM were elastic. It was later realized , in view of the shape of the shape of AGs  at late times, that plastic collisions - wherein the intercepted  ISM is engulfed  by the CB - was a better approximation in the AG phase.
  4. Figure 16. The prediction of the CB model for a constant density, \alpha=\beta+1/2,  becomes quite accurate  once the swept in  relativistic ISM mass in the CB rest frame  becomes  larger compared its rest mass.  It is satisfied  by far better than the FB closure relation for all the GRBs with very accurate measured values of \alpha and  \beta  in GRBs with a very bright X-AG which was followed  accurately up  to a very long time after burst.

Reviewer 3 Report

The authors presented a comprehensive comparative analysis for two physical models in gamma-ray bursts, and concluded that their cannonballs (CBs) model could be a better model to consistent with the observations than the fireball model (FB). This paper can be accepted after minor revision. 
My major point is that, for several critical tests, some clear evidences that the CBs is against the FB are not presented, such as the test #2, #3, #9.
Minor points:
1> Line 2: gamma ray bursts --> gamma-ray bursts
2> Table 1: some wrong typos
3> some sections are wrong indexed, such as section 7 and 8, which should be presented as a whole section.

Author Response

We thank the referee for his comments , however,

The referee remarks that "several critical tests do not compare the CB model PREDICTIONS  against the FB model PREDICTIONS such as tests #2, #3 #9.  But as stated clearly in our review we limited our comparison to falsifiable predictions.  In particular at the end of test #2 we stated  "The (CB predicted) correlations  just discussed, snugly satisfied by the data "  But the FB models  have not been  shown TO PREDICT OR EXPLAIN THE AMATI CORRELATION, NEITHER FOR GRBs, NOR FOR SHBs.  At the end of test #3 there is an  entire  sub-sub section (3.3.2) explaining how poorly the FB models deal with pulse shape. In spite of this, we have now added to the last paragraph (with   a new reference, now [43] ):  NO UNIVERSAL SHAPE OF THE GRB AND SHB PROMPT EMISSION PULSES CONSISTENT WITH THEIR TEMPORAL AND SPECTRAL BEHAVIOR HAS BEEN PUBLISHED. See., e.g., the FB model's predicted shape in Fig. 1  of  \cite{KPS}.   We had refrained from citing references such as this, because of being pathetical. Test #9:  We do not know of any claim based on the FB model which explains the universal behavior of the  afterglow of SN-less GRBs (which in the FB model are produced by collapsars - direct collapse of a massive star to a black hole without a supernova.

The "minor points" 1 and 2 have been dealt with. Minor point 3 has been dealt with by eliminating subsection 7.1.  & and 8 continue to have their own numbering and content. They deral with different subjects: The progenitors of GRBs, The progenitors of SHBs.  Note that not only the titles, but also the progenitors  are different. The single Typo in Table I has been corrected.